# Efficient and accurate frailty model approach for genome-wide survival association analysis in large-scale biobanks

Rounak Dey [1,9], Wei Zhou [2,3,4,5,9], Tuomo Kiiskinen [5,6], Aki Havulinna[5,6], Amanda Elliott[1,2,3], Juha Karjalainen[2,3,4,5], Mitja Kurki[2,3,4,5], Ashley Qin[1], FinnGen*, Seunggeun Lee [7], Aarno Palotie [2,3,4,5], Benjamin Neale [2,3,4,10], Mark Daly[2,3,4,5,10] & Xihong Lin [1,3,8] ✉

With decades of electronic health records linked to genetic data, large bio-banks provide unprecedented opportunities for systematically understanding the genetics of the natural history of complex diseases. Genome-wide survival association analysis can identify genetic variants associated with ages of onset, disease progression and lifespan. We propose an efficient and accurate frailty model approach for genome-wide survival association analysis of censored time-to-event (TTE) phenotypes by accounting for both population structure and relatedness. Our method utilizes state-of-the-art optimization strategies to reduce the computational cost. The saddlepoint approximation is used to allow for analysis of heavily censored phenotypes (>90%) and low frequency variants (down to minor allele count 20). We demonstrate the performance of our method through extensive simulation studies and analysis of five TTE phenotypes, including lifespan, with heavy censoring rates (90.9% to 99.8%) on ~400,000 UK Biobank participants with white British ancestry and ~180,000 individuals in FinnGen. We further analyzed 871 TTE phenotypes in the UK Biobank and presented the genome-wide scale phenome-wide association results with the PheWeb browser.

Survival models, especially the Cox proportional hazard model[1], have been widely used to analyze time-to-event (TTE) outcomes, both in biomedical research[2–4] and in genome-wide association studies (GWAS)[5–11]. It has been shown that the proportional hazard model can increase the power to detect genetic variants associated with the age-of-onset of TTE phenotypes in cohort studies compared to modeling the disease status using a logistic regression model, especially for common events[12–14]. Studying the genetic underpinning of age-of-onset, eg. early or late age-of-onset, is of substantial interest for

understanding the disease etiology and planning interventions. With the availability of detailed time-stamped diagnosis data from Electronic Health Records (EHR), large biobanks, such as UK Biobank (UKBB)[15] (>400,000 individuals) and FinnGen (https://www.finngen.fi/en) (>200,000 individuals), provide unprecedented opportunities to analyze TTE phenotypes to unravel the complex genetic architectures of disease onset, progression, and lifespan. Genome-wide scans of TTE phenotypes in large biobanks can potentially identify novel genetic variants associated with the onset of human diseases

[1]Department of Biostatistics, Harvard T.H. Chan School of Public Health, Boston, MA 02115, USA. [2]Analytic and Translational Genetics Unit, Massachusetts General Hospital, Boston, MA, USA. [3]Program in Medical and Population Genetics, Broad Institute of Harvard and MIT, Cambridge, MA, USA. [4]Stanley Center for Psychiatric Research, Broad Institute of Harvard and MIT, Cambridge, MA, USA. [5]Institute for Molecular Medicine Finland, Helsinki Institute of Life Sciences, University of Helsinki, Helsinki, Finland. [6]Finnish Institute for Health and Welfare, Helsinki, Finland. [7]Graduate School of Data Science, Seoul National University, Seoul, Korea. [8]Department of Statistics, Harvard University, Cambridge, MA, USA. [9]These authors contributed equally: Rounak Dey, Wei Zhou. [10]These authors jointly supervised this work: Benjamin Neale, Mark Daly, Xihong Lin. *A list of authors and their affiliations appears at the end of the paper. ✉e-mail: xlin@hsph.harvard.edu

by leveraging both the disease status and the age-of-onset information.

In GWAS analysis, population structure and sample relatedness are often key confounders and factors that need to be controlled for. Biobank cohorts often have substantial population structure and relatedness. For example, in the UK Biobank, 91,392 out of 408,582 subjects with White British ancestry have at least one relative (up to 3rd degree) in the data. Several methods based on linear[16–18] and logistic[19,20] mixed effects models have been developed to account for relatedness in GWASs for quantitative and binary phenotypes. To account for related subjects in the proportional hazard model, frailty models, which are mixed effects survival models, have been proposed[21,22], where event times are assumed to be independent conditional on unobserved random effects called "frailties". The frailties are modeled based on the dependence and clustering structure of the observations.

Previous research has extensively studied shared frailty models with Gamma-distributed frailties[21,23–28]. However, the shared frailty model assumes that the subjects in a cluster share common frailty and thus is limited in its scope to model more complicated dependency structures that arise in cohort-based association studies. Bivariate extensions to the shared frailty model such as the correlated Gamma[29,30] or the correlated compound Poisson[31] frailty model allow the frailties to be correlated among two subjects. However, these models are also too restrictive because they model the correlations using one parameter, and effectively, they are more appropriate for twin studies, and cannot model arbitrarily complex relationship structures.

To model complicated dependency structures, such as known familial structures and cryptic relatedness, the multivariate frailty model with Gaussian frailty was proposed[32,33], and was later implemented in the R package COXME[34], which, however, lacks scalability for GWASs. Recently the COXME method was further improved in COXMEG[35], which utilizes several computational optimization strategies to make it applicable in genetic association studies, but COXMEG still cannot handle biobank-scale genome-wide datasets. Based on our performance benchmarking, for 20,000 subjects, COXMEG requires 3356 CPU-hours (1412 CPU-hours for the COXMEG-Sparse option) to perform a GWAS of 46 million variants, thus COXMEG would take over 4.6 days (1.9 days for COXMEG-Sparse) to complete the GWAS, even with perfect parallelization on 30 CPUs.

In large-scale GWASs, the score test is particularly useful among different asymptotic tests, because it requires fitting the model only once under the null hypothesis of no association[20]. Score tests have also been implemented in COXMEG[36]. However, score tests can lead to severe type I error inflation for phenotypes with heavy censoring, where the number of subjects who have experienced an event (for example, diagnosed with the phenotype of interest) is small compared to the number of subjects who have not experienced the event (also called censored subjects) during the study follow-up period. This is common in biobank-based phenotypes. In the UK Biobank phenome that was built based on Phecodes[37] (see the "Methods" section), 871 TTE phenotypes have at least 500 events (cases), out of which 811 phenotypes have a censoring rate of more than 95%. The inaccuracies of the score test in unbalanced case-control phenotypes have been previously shown for logistic regression and logistic mixed effects models[19,38–40], and a saddlepoint approximation[41] (SPA)-based adjustment has been proposed and successfully implemented[19] to accurately calibrate the $p$-values in such scenarios. Recently, the SPACox[11] method also used SPA to calibrate $p$-values for time-to-event phenotypes in unrelated samples. However, the SPACox method does not account for sample relatedness. Through simulations, we show similar inaccuracies are also present in score tests in frailty models for analyzing heavily censored phenotypes.

Here, we propose a novel method for genome-wide survival analysis of TTE phenotypes, which accounts for both population structure and sample relatedness, controls type I error rates even for phenotypes with extremely heavy censoring, and is scalable for genome-wide scale phenome-wide association studies (PheWASs) on biobank-scale data. Our method, Genetic Analysis of Time-to-Event phenotypes (GATE), transforms the likelihood of a multivariate Gaussian frailty model into a modified Poisson generalized linear mixed model (GLMM[20,42]) likelihood, employs several state-of-the-art optimization techniques to fit the modified GLMM under the null hypothesis, and then performs score tests calculated using the null model for each genetic variant. To obtain well-calibrated $p$-values for heavily censored phenotypes, GATE uses the SPA to estimate the null distribution of the score statistic instead of the traditionally used normal approximation. Moreover, our method saves the memory requirement substantially by storing the raw genotypes in binary format and calculating the elements of the GRM on the fly instead of storing or inverting a large dimensional GRM. Through extensive simulations and analysis of TTE phenotypes from the UK Biobank data of 408,582 subjects with White British ancestry as well as the FinnGen study freeze 5 that contains 218,792 subjects, we showed that GATE is scalable to biobank-scale GWASs of TTE phenotypes with type I error rates well controlled even for less frequent variants and heavily censored phenotypes. Benchmarking has shown that GATE can analyze 46 million variants in a GWAS with 408,582 subjects in ~14.5 h using 30 CPUs with peak memory usage under 11 GB.

## Results
### Overview of methods
GATE consists of two main steps: (1) Fitting the null frailty model to estimate the variance component and other model parameters and (2) performing a score statistic-based test for association between each genetic variant and the phenotype. Step 1 involves iteratively fitting the null frailty model by first rewriting the likelihood of the observed censored time to event data under the frailty model as a modified Poisson log-linear mixed effects model likelihood, and then applying modified optimization strategies as described in GMMAT[20] and SAIGE[19] to fit the null modified Poisson log-linear mixed models (METHODS). They include using the computationally efficient average information restricted maximum likelihood (AI-REML[20,43]) algorithm for estimating the variance component and using the pre-conditioned gradient descent (PCG[44]) method to solve linear systems to avoid inverting the $N \times N$ genetic relatedness matrix (GRM), where $N$ is the number of subjects. GATE computes the elements of the GRM on-the-fly when needed using binary vectors of raw genotypes, and thus it does not require supplying, storing, or inverting a pre-computed GRM, which can be extremely time and memory-consuming for large sample sizes ($N$). For example, in the UK Biobank data with $M = 93,511$ markers and $N = 408,582$ subjects with White British ancestry, the memory requirement drops from 622 GB for storing a pre-computed GRM in floating point numbers, to only 8.9 GB for storing the raw genotypes in the binary format.

Step 2 involves scanning the entire genome and testing each variant for association using score statistics. Since the overall cost of computing the variance of the score statistic for all variants is extremely high because it involves operations on the large-dimensional GRM, in step 2, GATE uses a variance ratio approximation derived under the modified TTE Poisson loglinear models by extending that used in existing LMM and GLMM-based methods such as GRAMMAR-Gamma[17], BOLT-LMM[16], fastGWA[18], and SAIGE[19]. The ratio of the variance of the score statistic with and without random effects (and an attenuation factor due to estimating the baseline hazards) is computed using a subset of genetic markers. Previously, it was shown that this variance ratio remains approximately constant for variants with MAC ≥ 20 for LMM and GLMMs. Through analytical derivations and simulation examples, we show this observation holds for frailty models as well (Supplementary Note section 3 and Supplementary Fig. 15). Therefore, when performing the genome-wide scan, the variance of

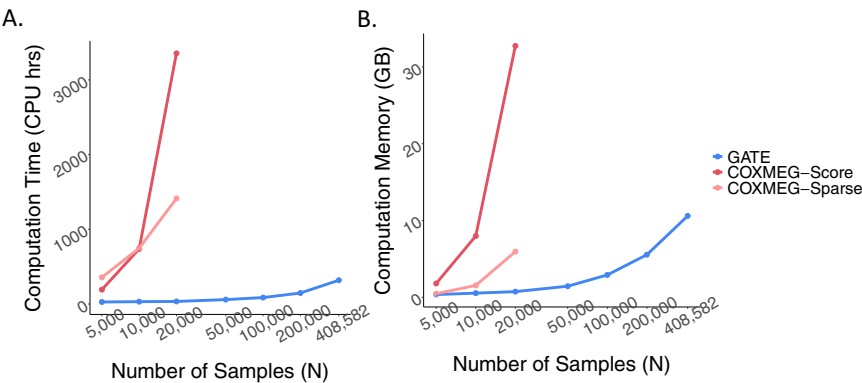

**Fig. 1 | Projected computation cost for GATE, COXMEG-Score, and COXMEG-sparse as a function of sample size. A** is for computation time and **B** is for memory usage. The numerical data are provided in Supplementary Table 1. Benchmarking was performed for the GWAS of lifespan based on randomly sub-sampled data from UK Biobank White British ancestry subjects. Association tests were performed on 200,000 randomly selected markers with imputation INFO ≥ 0.3, with the filtering criteria of MAC ≥ 20. The computation times were projected for testing 46 million variants with INFO ≥ 0.3 and MAC ≥ 20. The reported run times are medians of five runs, each with randomly sampled subjects with different randomization seeds. The *x*-axis is plotted on a log10 scale.

the score statistic is computed without using the GRM and then calibrated using the variance ratio.

Next, GATE uses the saddlepoint approximation[41] (SPA) to approximate the null distribution of score statistics for association tests under the modified Poisson log-linear mixed models. SPA-based tests have been successfully used for logistic regression[39] and logistic mixed models[19] and provide more accurate *p*-values than traditional score tests under normal approximation for low-frequency variants when the case-control ratio is unbalanced. In GATE, we have implemented an efficient SPA-based test for frailty models by extending the fastSPA method in Dey et al.[39]. Through simulations and real data analysis, we show that SPA tests provide accurate and calibrated *p*-values, even for low-frequency variants when the censoring rate is high to 99%.

Both GATE and COXMEG[35] conduct genetic association tests for TTE phenotypes using the frailty model. Besides the use of SPA-based tests, GATE uses the variance ratio approach to approximate the variances of the score statistics, while COXMEG calculates the variances using the GRM. Using simulation studies, we have shown that GATE provides association *p*-values consistent with COXMEG ($R^2$ of $-\log_{10}$ *p*-values > 0.99, slope = 1.008, intercept = −0.0004) for common variants (MAF > 5%) when the censoring rate is 50% and moderate sample sizes (Supplementary Fig. 1A). Further, GATE has well-controlled type I error rates even for less frequent variants and phenotypes with heavy censoring rates, where COXMEG results in inflated type I error rates (Supplementary Fig. 1B). Further, as shown below, GATE is computationally much more scalable than COXMEG for large biobank data.

**Computation and memory costs**

To assess the computational performance of GATE and the tests implemented in the COXMEG package, namely COXMEG-Score and COXMEG-Sparse, we randomly sampled subsets of different sample sizes from 408,582 UK Biobank subjects with White British ancestry. We then benchmarked association tests for overall lifespan (16,375 events, 389,721 censored) adjusting for the top four ancestry principal components, birth year, and sex using GATE, COXMEG-Score, and COXMEG-Sparse on 200,000 variants randomly selected from 46 million genetic variants with imputation info ≥0.3 and MAC ≥ 20. In Step 1, 93,511 high-quality genotyped markers were used for the GRM. The projected overall computation time (Fig. 1 and Supplementary Table 1) for GATE to analyze 46 million variants on *N* = 408,582 subjects was 318 CPU-hours, and the actual computation time on a machine with 30 cores was 14.5 h. Step 2, which accounts for the majority of the

computation time (95.4% for *N* = 408,582) requires substantially less memory (peak memory usage 0.85 GB) than Step 1 (peak memory usage 10.6 GB).

However, to perform GWAS on only 20,000 subjects, the projected computation time and memory usage for COXMEG-Score were 3356 CPU-hours (4.6 days with 30 CPUs) and 32.75 GB, respectively, and for COXMEG-Sparse, they were 1412 CPU-hours (1.96 days with 30 CPUs) and 5.95 GB. As GATE only uses 34 CPU-hours and 0.74 GB, it achieves 98% and 88% reductions in computation time and memory, respectively, compared to COXMEG. Note that the computation time and memory requirements increase nearly linearly with the sample size for GATE, whereas they increase quadratically for COXMEG-Score and COXMEG-Sparse.

**Phenome-wide GWAS of time-to-event phenotypes in the UK Biobank data**

We have applied GATE to perform phenome-wide GWAS for 871 UKBB TTE phenotypes with at least 500 events, adjusting for the top four PCs, birth year, and sex (except for 93 sex-specific phenotypes). The TTE phenotypes were created based on the International Classification of Disease (ICD) codes version 9 and 10 mapped to the PheWAS code (PheCode[37]) definitions (see the "Methods" section) as well as their associated diagnosis dates in the UK Biobank electronic medical records. For each phenotype, we analyzed approximately 46 million genetic markers imputed from the Haplotype Reference Consortium[45] panel and UK10K[46] with imputation INFO score ≥ 0.3 and MAC ≥ 20. Among the 408,582 UK Biobank subjects with White British ancestry, 91,392 had at least one relative up to a third degree[15]. To account for the relatedness among the subjects, we used 93,511 high-quality genotyped markers with MAF ≥ 0.01 to construct the GRM in Step 1. The same set of markers was used by the UK Biobank research group[15] for estimating kinship among the samples because they are only weakly informative of the ancestry and therefore provide more accurate kinship estimates. We also performed a sensitivity analysis using a larger set of markers (245,745) for the four exemplary phenotypes discussed before (see Supplementary Note Section 7). We further applied SPA-based adjustment of the score test because the censoring rates (Supplementary Fig. 2) were extremely high for most of the TTE phenotypes in the UKBB (for example, 811 out of 871 have a censoring rate of more than 95%). The summary statistics for all 871 PheCodes analyzed using GATE are available to download from a public repository (see the section "Data availability") and browsed in the PheWeb[47] (see the section "Data availability").

Here we discuss the association results using four phenotypes with different censoring rates as exemplars: ischemic heart disease (IHD: PheCode 411, $N$ events = 36,962, $N$ censored = 370,814, censoring rate = 90.9%), female breast cancer (FBC, PheCode 174.1, $N$ events = 15,396, $N$ censored = 192,764, censoring rate = 92.6%), glaucoma (PheCode 365, $N$ events = 6046, $N$ censored = 392,925, censoring rate = 98.5%), and Alzheimer's Disease (AD: PheCode 290.11, $N$ events = 822, $N$ censored = 342,059, censoring rate = 99.8%). The Manhattan and QQ plots for the GWAS of these phenotypes using GATE with and without SPA are presented in Figs. 2 and 3, respectively. The results demonstrate that not adjusting for SPA greatly inflates the type I errors, especially for the low-frequency variants, whereas the SPA-adjusted method shows well-controlled type I error rates. In total, 114 loci have been identified for the four TTE phenotypes: 55 for IHD, 37 for FBC, 19 for glaucoma, and 3 for AD. We also applied GATE to these four phenotypes in the FinnGen study (see the "Methods" section) and 81 out of the 114 loci were also tested in the FinnGen study, of which 78 had the same effect direction in both UKBB and FinnGen. 69 out of the 81 loci were successfully replicated in FinnGen with $p$-value < 0.05. The complete list of all significant loci and the association results in the UKBB, FinnGen as well as the meta-analysis of the two data sets are reported in Supplementary Data 1. Overall, 99 out of the 114 significant loci have been previously reported to be associated with disease risk in case-control studies to the best of our knowledge. Several loci that are previously well known as associated with the risk of the diseases have been identified in our study, such as the loci *LPA* and *CELSR2* for IHD[48,49], *FGFR2*[50] and *CASC16*[51] for breast cancer, *MYOC*[52] and *TMCO1*[53] for glaucoma, and *APOE* e4 variant for AD[54]. The age-varying predicted risk of disease onset based on the GATE method, and the age-varying disease-free probability by genotypes based on the Kaplan–Meier curve[55] for the exemplary top hits was plotted in Fig. 4 and Supplementary Fig. 3, respectively.

We further applied logistic mixed models using SAIGE to analyze these four UKBB phenotypes using their binary disease status at the latest follow-up time, accounting for the same covariates as in the GATE application. GATE identified 18 loci (11 for IHD, 2 for FBC, 4 for glaucoma, and 1 for AD) that were not significant using SAIGE logistic mixed models (see Supplementary Data 2). Out of these 18 loci, 12 were previously reported as associated with the corresponding phenotypes in other case-control studies. For example, GATE identified an association between AD and an intronic rare variant rs533100590 (MAF = 0.005%, $p$-value = $2.78 \times 10^{-8}$) in gene *ATP9B* (ATPase, class II, type 9B) while SAIGE did not ($p$-value = $1.09 \times 10^{-6}$). This locus has been previously shown to be associated with AD[56]. GATE identified the known locus *SWAP70* (intronic rs378825, MAF = 42.7%, $p$-value = $4.92 \times 10^{-8}$) for IHD that was missed by SAIGE logistic mixed model ($p$-value = $1.38 \times 10^{-7}$).

## GWAS of lifespan in the FinnGen study and the UK Biobank

We have also applied GATE to the overall lifespan in the FinnGen study ($N$ events = 15,152, $N$ censored = 203,244), in which the age of death ranges from 7 years old to 106 years old as shown in Supplementary Fig. 4. We identified the previously reported *APOE* locus for lifespan[57] in FinnGen, in which the most significant variant is the APOE-e4 missense variant rs429358 (MAF = 18.3%, $p$-value = $1.01 \times 10^{-14}$) and it is well-known to be associated with lifespan, cardiovascular diseases, stroke, and Alzheimer's disease[58–60]. However, when SAIGE logistic mixed model was applied to the Finngen binary trait of dead/alive status, it did not identify any significant locus (rs429358 has $p$-value $1.89 \times 10^{-6}$).

The locus rs429358 has also been replicated in UKBB ($N$ events = 16,375 and $N$ censored = 389,721, see Supplementary Fig. 5A) with $p$-value $1.92 \times 10^{-5}$ and meta-analysis $p$-value $4.04 \times 10^{-17}$ (Supplementary Table 2 and Supplementary Fig. 5B).

The top hit in UKBB for lifespan (rs157592, MAF = 18.7%, $p$-value = $1.87 \times 10^{-8}$) had LD $r^2 = 0.7$ with rs429358 as presented in the Supplementary Table 2. This variant rs157592 is in the intergenic region and has no in-silico function according to the FAVOR functional annotation online portal[61] (see the section "Code availability").

## Simulation studies

We investigated the type I error rates and power of GATE in the presence of sample relatedness using 10,000 simulated samples. Due to computational burden, we used GATE-noSPA instead of COXMEG-Score for type I error evaluation as Supplementary Fig. 1C shows the two approaches provide consistent association $p$-values ($R^2$ of $-\log10$ $p$-values > 0.99).

The type I error rates of GATE was evaluated based on association tests of $9.4 \times 10^8$ simulated genetic markers on 10,000 samples, which contain 500 families and 5000 independent samples. Each family has 10 members, simulated based on the pedigree shown in Supplementary Fig. 6. The variance component parameter $\tau$ is set to be 0.1 and 0.25 (see the "Methods" section). The empirical type I error rates at the significance level $\alpha = 1 \times 10^{-6}$ and $5 \times 10^{-8}$ are shown in Supplementary Table 3 and Supplementary Fig. 7A. Our simulation results suggest that GATE has well-controlled type I error rates even for low-frequency variants (down to MAC = 20) when the phenotype is heavily censored (90%). However, without SPA, the score tests in GATE suffer from inflated type I error rates as the censoring becomes more extreme and the frequency of variants decreases. We also evaluated type I error rates of GATE in a setting with cryptic sample relatedness by randomly selecting 10,000 UKBB participants with white British ancestry. Phenotypes were simulated using the real genotypes in the UKBB to mimic the sample relatedness of a real-world dataset, and association tests were conducted on the imputed genetic markers in the UKBB (see the "Methods" section). Similarly, we observed that the type I error rates were well controlled in GATE in presence of cryptic sample relatedness with different censoring rates (Supplementary Table 4, Supplementary Figs. 7B and 8).

Next, we evaluated the empirical power of GATE at $\alpha = 5 \times 10^{-8}$ and compared it to the power of COXMEG-Score. Supplementary Fig. 9 shows the power curve by hazard ratios for variants with MAF 0.05 and 0.2 when $\tau = 0.25$ and the censoring rate = 50%. Both methods have nearly identical power in all simulation settings. We do not compare their powers in the presence of heavy censoring, in view of the inflated type I error rate of COXMEG-Score.

Overall simulation studies show that GATE can control type I error rates even when the censoring rate is high and has similar power for common variants as COXMEG-Score. In contrast, same as GATE-noSPA, COXMEG suffers type I error inflation and the inflation is especially severe with low MAF and heavy censoring (Supplementary Figs. 1B, C, 7, and 8).

In addition, we compared the empirical power of GATE and the association tests based on a logistic mixed model as implemented in SAIGE (Supplementary Fig. 10), for simulated TTE phenotypes with 50%, 75%, and 95% censoring rates (see the "Methods" section). SAIGE treated all events at the latest follow-up time as cases and all censored individuals as controls, and tested for associations between genetic markers and the disease risk coded as the case-control status while accounting for the age at the latest follow-up time as a covariate in the linear term. As expected, GATE overall showed a higher empirical power to identify the genetic markers that are associated with the phenotype than SAIGE. The difference in the empirical powers decreased as the censoring rate increased. However, even for the datasets with a 95% censoring rate, GATE empirically had ~5–6% power improvement over SAIGE at the hazard ratio range 2–3 for MAF 0.05, and at the hazard ratio range 1.5–1.8 for MAF 0.2.

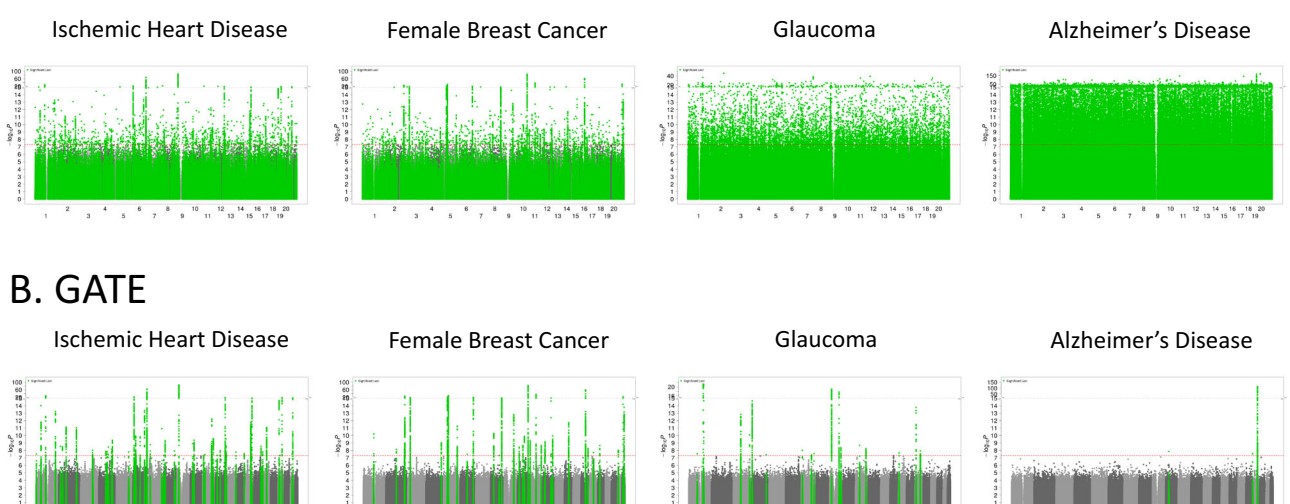

**Fig. 2 | Manhattan plots for GWAS of four time-to-event phenotypes with different censoring rates in the UK Biobank data with White British ancestry.** GWAS results using GATE-noSPA (**A**) and GATE (**B**) are shown for ischemic heart disease (PheCode 411, $N = 407{,}776$, censoring rate = 90.9%), female breast Cancer (PheCode 174.1, $N = 208{,}160$, censoring rate = 92.6%), glaucoma (PheCode 365, $N = 398{,}971$, censoring rate = 98.5%), and Alzheimer's Disease (PheCode 290.11, $N = 342{,}881$, censoring rate = 99.8%).

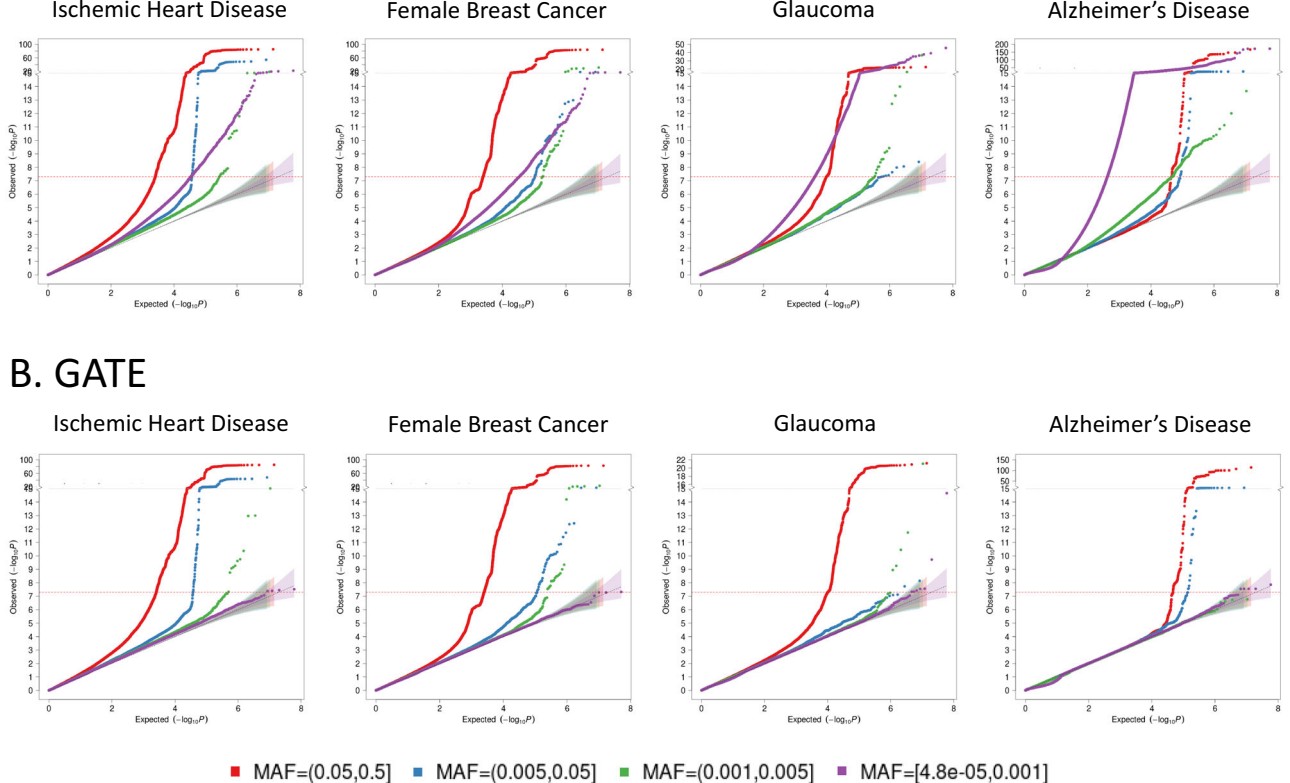

**Fig. 3 | Quantile–quantile (QQ) plots for GWAS of four time-to-event phenotypes with different censoring rates in the UK Biobank data with White British ancestry.** GWAS results using GATE-noSPA (**A**) and GATE (**B**) are shown for ischemic heart disease (PheCode 411, $N = 407{,}776$, censoring rate = 90.9%), female breast Cancer (PheCode 174.1, $N = 208{,}160$, censoring rate = 92.6%), glaucoma (PheCode 365, $N = 398{,}971$, censoring rate = 98.5%), and Alzheimer's Disease (PheCode 290.11, $N = 342{,}881$, censoring rate = 99.8%). QQ plots are color-coded based on different minor allele frequency categories. 95% error bands around the nominal $x = y$ diagonal line are also shown for each MAF category.

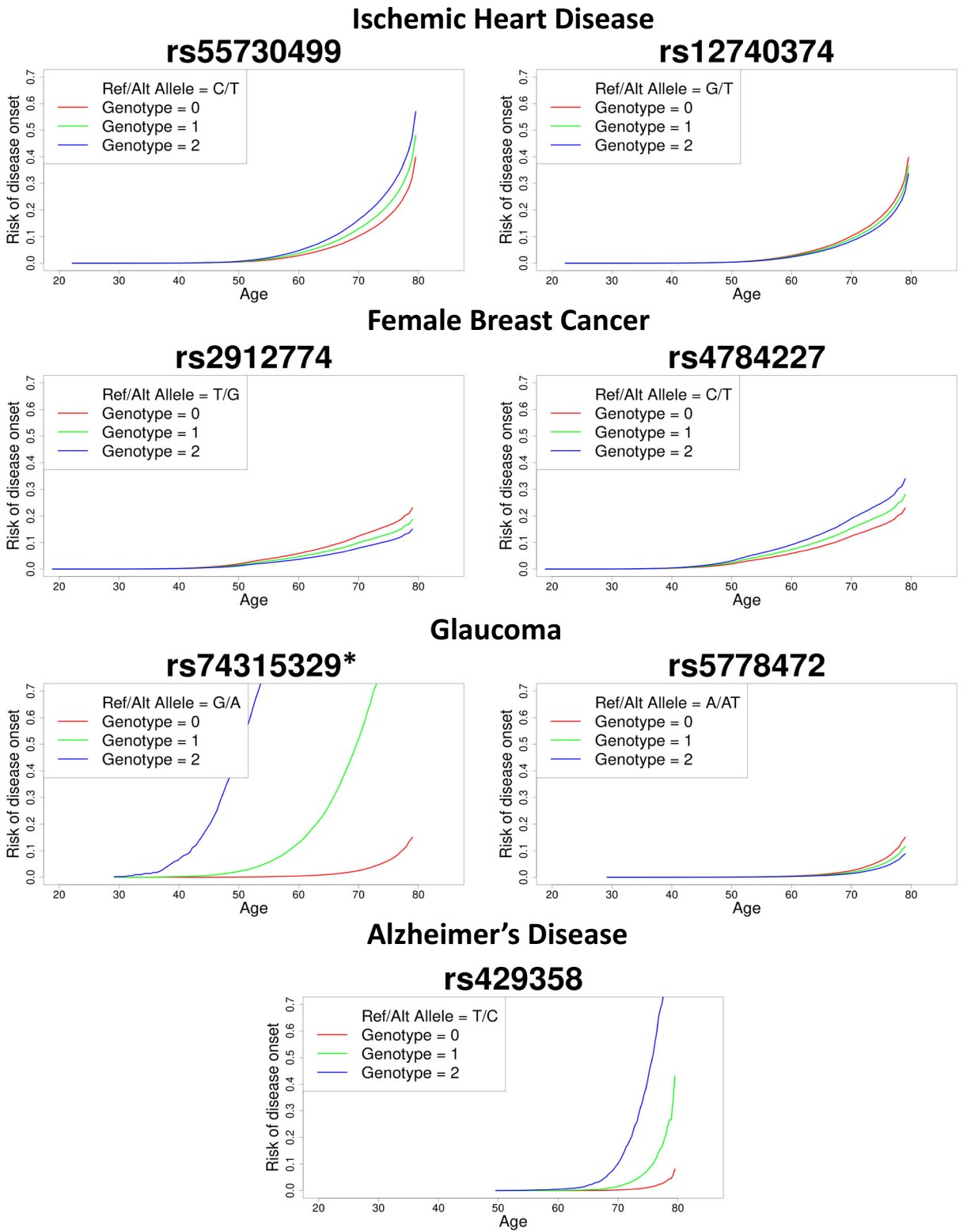

* **No homozygous alternate subject was present among the study subjects for rs74315329 (Alternate allele frequency = 0.0013)**

**Fig. 4 | Predicted risk of disease onset over age for the top two loci in the GWAS of four phenotypes in the UK Biobank data with White British ancestry.** Predicted risk of disease onset is plotted over age by genotypes for loci LPA and CELSR2 for ischemic heart disease, FGFR2 and CASC16 for female breast cancer, MYOC and TMCO1 for glaucoma, and APOE e4 variant for AD. The red, green, and blue lines represent the risk of disease onset for alternate allele counts zero, one, and two, respectively, for a female subject born in 1950 (median birth year in the UKBB data) with the top four PC coordinates each set at the mean level across the UK Biobank subjects with white British ancestry.

## Discussion

In this paper, we have proposed a novel method to perform scalable genome-wide survival association analysis of censored TTE phenotypes in biobank-scale data using an efficient implementation of the frailty model. Our method can adjust for population structure and sample relatedness and provide accurate $p$-values even in extreme cases of very low-frequency variants and heavily censored phenotypes (incidence rate < 0.1%). Applying this approach to the UK Biobank and the FinnGen study, we demonstrated that our method is scalable to the analysis of large biobank-scale datasets with >400,000 subjects.

The major methodological improvement in our study is to derive the frailty model as a modified Poisson log-linear mixed model, which allows us to incorporate some of the existing approaches for GLMM-based models into the frailty model for time-to-event (TTE) phenotypes. Frailty model is entirely different from a GLMM, both in the kind of data they are used to analyze and the problems they are applied to, as well as in the ways these models fit. In our paper, we derived and transformed the frailty model likelihood into a modified Poisson GLMM likelihood (Supplementary Note Section 1), and derived the appropriate model fitting procedure for such modified GLMMs. We note that the modified Poisson GLMM is still not a GLMM, and thus the derivation steps and model-fitting techniques are non-trivial.

Biobanks with genetic data linked to EHR records/survey questionnaires provide unprecedented opportunities for genetic association studies on TTE phenotypes to identify genetic risk factors that affect the onset and progression of diseases. However, biobanks pose challenges to such analysis because of the high computational and memory cost required to handle large data sets with extensive population structure and relatedness. Moreover, existing methods such as COXMEG, artificially inflate associations when heavily censored phenotypes (e.g., censoring rate > 75%) and low-frequency variants (MAF < 1%) are involved. The proposed method GATE performs a frailty model-based association analysis to account for both population structure and relatedness using score tests with SPA adjustment, which provides accurate $p$-values under heavy censoring. In addition, it implements several optimization techniques that were previously used in the context of linear and logistic mixed models in BOLT-LMM and SAIGE to make it computationally feasible to analyze large biobank cohorts. We have applied GATE to 871 TTE phenotypes in the UK Biobank data with White British ancestry, which were constructed based on PheCodes mapped to ICD codes and have at least 500 events. The genome-wide summary statistics are available for the public to download. We have also created a PheWeb[47] for users to explore and visualize the PheWAS results.

Lifespan is a typical TTE phenotype and the genetic effects on lifespan can be appropriately modeled by the frailty model. We applied GATE to lifespan in the FinnGen study, whose participants have a wide age-of-death range from 7 to 106 years, and have successfully identified a well-known locus rs429358 ($p$-value = $1.01 \times 10^{-14}$). However, this locus has been missed by a logistic mixed model for the dead/alive status. ($p$-value = $1.89 \times 10^{-6}$). This example suggests that applying frailty models can be useful for uncovering genetic risk factors for TTE analysis, as further evidenced through simulation studies (see the "Methods" section). GATE can facilitate these studies.

We also compared GWAS results using logistic mixed models of the binary disease status as implemented in SAIGE in the four example phenotypes presented in the paper and found that across the four phenotypes, SAIGE failed to identify 18 loci (Supplementary Data 2) that were identified GATE, among which 11 were for ischemic heart disease. This shows that a frailty model-based analysis of TTE phenotypes can lead to the identification of loci that might be missed by only analyzing the disease status using a logistic mixed effects model. The scatter plots comparing the association $p$-values from GATE and SAIGE

(Supplementary Fig. 11) show that for ischemic heart disease and glaucoma, the $p$-values based on GATE overall tend to be smaller than SAIGE, and for female breast cancer and Alzheimer's disease, the $p$-values are similar between these two methods. The TTE outcome is different from the binary case-control outcome, and logistic models can result in loss of power for such outcomes, especially for common events. Although the TTE phenotypes in biobanks such as the UK Biobank and FinnGen are currently subject to heavy censoring, as the biobank participants are followed over time, more events will be observed. As events will become more common over time in biobank follow-up, the power gain of GWAS analysis of TTE phenotypes using frailty models via GATE over logistic mixed models via SAIGE will increase. Logistic models with age (at disease onset or at the latest follow-up time) as a covariate assume a homogenous effect (in logit scale) of age on the risk of the disease, which may not be valid, especially when the definition of the age covariate can be different between the cases/failure events (age-of-onset) and the controls/censored (age at the latest follow-up time). Survival models, on the other hand, are developed specifically to accommodate age-of-onset and age-of-censoring differently, and they model the effect of age on the disease-risk non-parametrically in the baseline hazard without the homogeneity assumption.

TTE phenotypes are particularly suited not only for studying disease onsets but also for exploring other progression phenotypes such as times of surgery, recurrence, times of onset of secondary phenotypes after an initial diagnosis, etc. Previously, the lack of scalable GWAS methods for TTE outcomes hindered such investigations on massive scales. By facilitating large-scale GWAS of TTE phenotypes, GATE opens the door to such investigations in the future at genome-wide and phenome-wide scales. Further, modeling TTE phenotypes also has the added advantage of designing appropriate intervention responses. Since frailty models explicitly model the age-of-onset of the disease, one can design interventions based on the genetic predispositions of the subjects, and also based on whether the disease has early or late onset. Logistic models are not particularly suitable for this purpose as it models the effect of age as a homogeneous effect, which is a much stronger assumption compared to the non-parametric modeling of age-of-onset in survival models.

One consideration while analyzing TTE phenotypes is the appropriate choice of unit of time. To assess the impact of time-units on the GWAS results, we performed a sensitivity analysis using the event and censoring times rounded to the nearest 1 month, 3 months, 6 months, and 12-month time-units for the four exemplary UK Biobank phenotypes presented in this paper, and compared the $p$-values across different time-units (Supplementary Fig. 12). The $p$-values were very similar across the four time-units for all phenotypes, with more detailed time-units resulting in slightly more significant $p$-values.

For the selection of a number of markers to construct the GRM, there is a trade-off between computation cost and the accuracy of adjusting the sample relatedness. Increasing the number of markers ($M$) included in the GRM linearly increases the computation time and memory requirement of step 1, whereas using too few markers may not be sufficient to capture the detailed familial and cryptic relatedness among the samples properly[62]. For the UK Biobank data analysis, we used $M$ = 93,511 LD pruned high-quality genotyped markers which were used by the UK Biobank research group for estimating kinship among the samples[15]. We performed a sensitivity analysis (see Supplementary Note Section 7) by increasing the number of markers to $M$ = 245,975 pruned markers with MAF ≥ 0.01. The results (Supplementary Figs. 13 and 14) showed that the $p$-values were generally concordant, and the $p$-values using $M$ = 245,975 markers were slightly larger than the $p$-values using $M$ = 93,511 markers.

GATE has several limitations. First, similar to other mixed model methods for genetic association tests, the computation time required for the algorithms to converge in step 1 can vary among different

phenotypes and study samples because of the difference in heritability and the extent of sample relatedness. Second, to be computationally efficient, GATE uses a score statistic-based test without fitting the model under the alternate hypothesis. Therefore, it does not provide accurate estimates of hazard ratios for genetic variants. Following a similar approach as in several other mixed model-based methods[16,17,19,63], GATE provides hazard ratio estimates for genome-wide variants using the null model parameter estimates (see Supplementary Note Section 5). Alternatively, the GATE software also allows users to include variants one-at-a-time into the model for step 1 in order to get more accurate hazard ratio estimates. Third, GATE performs single-variant association analysis, which can suffer from low power to detect associations for rare variants. Significant single variant-based GWAS findings for rare variants need to be interpreted with caution, and replication of these findings using independent samples is important. To boost the power of rare variant association tests in whole genome/exome sequencing (WGS/WES) studies, set-based rare variant tests have been commonly used. It is of future research interest to extend GATE to mask-based or region-based rare variant set association tests in WGS/WES studies by extending burden, SKAT, and other tests[61,64] to frailty models for censored time-to-event data.

Fourth, the current version of GATE does not incorporate left-truncated data, which may not be valid for early-onset phenotypes in biobanks with relatively older participants. For example, the median age of UK Biobank's participants is 59 and the earliest dates of health data available are around the late 1990s. Assuming no left-censoring can reduce association power for early-onset diseases. Future work will extend GATE to accommodate left-truncated phenotypes. Fifth, since the follow-up information is based on EHR systems that record age-of-diagnoses instead of true age-of-onsets, the actual analysis presented in our paper is based on age-of-diagnoses. As long as age-of-diagnoses are close to age-of-onsets, analyzing them can be reasonable. However, as mentioned before, for left-truncated phenotypes, this may not always be the case. Specific care needs to be taken when analyzing such phenotypes. Finally, the frailty model presented in the paper assumes independent censoring, which is a common assumption in the survival analysis literature. However, for certain phenotypes like IHD, the event of death can be a "competing risk"[65–68] which may cause dependent censoring. Competing risk models generally involve other strong assumptions, for which we did not consider them in GATE which is intended to be applied under more general settings. In the future, we plan to include competing risk models into GATE as well for specific phenotypes which may have dependent censoring.

GWAS is an important first step of genetic discovery as evidenced by the extensive GWAS literature. The functions of many GWAS discoveries are unknown and there is a substantial need to identify causal functional variants of these GWAS disease-associated loci. Numerous large-scale efforts have been ongoing to study the functions of the variants identified by GWAS to accelerate discovery from genetic maps to biological mechanisms to physiology and medicine, and drug target discovery and prioritization. Examples include the recently launched NHGRI Impact of Genomic Variation on Function (IGVF) Consortium, Open Targets, and the International Common Disease Alliance (ICDA).

In summary, we have proposed a scalable and accurate method, GATE, to perform genome-wide PheWAS of TTE phenotypes on large biobank cohorts accounting for population structure, sample relatedness, and heavy censoring. We demonstrated that it is possible to efficiently analyze the current largest biobank (UK Biobank) of >400,000 subjects using GATE. Our method facilitates biobank-based PheWAS of TTE phenotypes which ultimately contributes towards identifying genetic components that affect the onset and progression of complex diseases.

## Methods

### Frailty model for Time-to-event phenotypes

Consider a study of $N$ subjects, where for the $i$th subject, we observe the data pair $(\delta_i, t_i)$, where $\delta_i$ is a censoring indicator, with $\delta_i = 1$ if the $i$th subject experiences an event during the study period, and $\delta_i = 0$ otherwise, i.e., censored. Let $t_i$ denote the observed event or censoring time. For the $i$th subject, let the $p \times 1$ vector $X_i$ denote the covariates, and $G_i = 0, 1, 2$ denote the minor allele counts for the genetic variant of interest. Then, in a frailty model[25,32,69], the conditional hazard function of subject $i$ at time $t$ given the covariates, genotype, and random effect/frailty $b_i$ is modeled as

$$\lambda_i(t|b_i) = \lambda_0(t)\exp(X_i^\top \beta + G_i \gamma + b_i) \tag{1}$$

where $\beta$ and $\gamma$ are the regression coefficients of the covariates $X_i$ and the genotype $G_i$ respectively, and $\lambda_0(t)$ is the baseline hazard function at time $t$, the frailty $b = (b_1, \ldots, b_N)$ follows a multivariate normal distribution $N(0, \tau V)$, with $V$ being the genetic related matrix (GRM). Unlike standard generalized linear mixed models, the covariate vector $X_i$ in a frailty model does not include the intercept term, instead, the baseline hazard $\lambda_0(t)$ works as the intercept in a frailty model. We test the null hypothesis of no genetic association $H_0 : \gamma = 0$ vs $H_1 : \gamma \neq 0$.

### Estimating the variance component and other null model parameters (step 1)

First, the likelihood for the observed event status–time pairs $(\delta_i, t_i)$ under the frailty model is derived and expressed as a modified Poisson mixed-effects model likelihood, with the mean function weighted by the cumulative baseline hazard (CBH) function $\Lambda_0(t) = \int_0^t \lambda_0(u)\mathrm{d}u$. The CBH function is estimated by the Breslow's estimator $\hat{\Lambda}_0(t)$ as a step function. Breslow[70] showed that the maximum likelihood approach for the proportional hazard model (for unrelated subjects) that leads to the estimator $\hat{\Lambda}_0(t)$, is equivalent to maximizing the partial likelihood proposed by Cox[1]. In Supplementary Note Section 6, we have shown that the same maximum likelihood approach holds for frailty models (related subjects) as well given the random effects. Then, using the penalized quasi-likelihood (PQL[42]) method and the AI-REML[43] algorithm, the model parameters under $H_0$ are estimated iteratively. To avoid storing large $N \times N$ GRMs, GATE only calculates the elements of the GRM when they are needed using raw binary format genotypes. For the scalable computation of quantities of the form $\mathbf{A}^{-1}x$ that arises in the model fitting steps, where $\mathbf{A}$ is a large matrix and $x$ is a vector, GATE uses the PCG algorithm[44], which has been previously used in BOLT-LMM[16] and SAIGE[19] to accurately compute quantities like $y = \mathbf{A}^{-1}x$ by solving the linear system of equations $\mathbf{A}y = x$, instead of explicitly inverting the large matrix $\mathbf{A}$.

Once the null model parameters, random effects, and cumulative baseline hazard functions $(\hat{\beta}, \hat{b}_i, \hat{\Lambda}_0(t_i))$ have been estimated, GATE estimates the variance ratio from a small number of markers. Denote the fitted means by $\hat{\mu}_i = \hat{\Lambda}_0(t_i)\exp(\mathbf{X}_i^\top \hat{\beta} + \hat{b}_i)$, and the weight matrix $\hat{W} = \mathrm{diag}(\hat{\mu}_1, \ldots, \hat{\mu}_N)$. Then the score statistic, under $H_0 : \gamma = 0$ is $T = \mathbf{G}^\top(\boldsymbol{\delta} - \hat{\boldsymbol{\mu}}) = \widetilde{\mathbf{G}}^\top(\boldsymbol{\delta} - \hat{\boldsymbol{\mu}})$, where $\mathbf{G} = (G_1, \ldots, G_N)$, $\boldsymbol{\delta} = (\delta_1, \ldots, \delta_N)$, $\hat{\boldsymbol{\mu}} = (\hat{\mu}_1, \ldots, \hat{\mu}_N)$. The covariate-and-intercept-adjusted genotypes are denoted by $\widetilde{\mathbf{G}} = \mathbf{G} - \widetilde{\mathbf{X}}(\widetilde{\mathbf{X}}^\top \hat{W}\widetilde{\mathbf{X}})^{-1}\widetilde{\mathbf{X}}\mathbf{G}$, where $\widetilde{\mathbf{X}} = [\mathbf{1X}]$ is the augmented covariate matrix. Then, the variance of the score statistic under $H_0$ is given by $\mathbf{V_T} = \mathbf{G}\hat{\mathbf{Q}}\mathbf{G} = \widetilde{\mathbf{G}}\hat{\mathbf{Q}}\widetilde{\mathbf{G}}$, where $\hat{\mathbf{Q}} = \hat{\mathbf{S}}^{-1} - \hat{\mathbf{S}}^{-1}\mathbf{X}(\mathbf{X}\hat{\mathbf{S}}^{-1}\mathbf{X})^{-1}\mathbf{X}\hat{\mathbf{S}}^{-1}$, $\hat{\mathbf{S}} = (\hat{\mathbf{W}} - \hat{\mathbf{U}})^{-1} + \hat{\tau}\mathbf{V}$. The expression of $\hat{\mathbf{U}}$ is described in detail in Supplementary Note Section 1.3. Unlike in the GLMMs, the term $\hat{\mathbf{U}}$ appears in the variance of the score statistic due to the attenuation of information (additional variability) for estimating $\Lambda_0(t_i)$s. The variance ratio is then calculated as $\hat{r} = \frac{\widetilde{\mathbf{G}}\hat{\mathbf{Q}}\widetilde{\mathbf{G}}}{\widetilde{\mathbf{G}}\mathbf{W}\widetilde{\mathbf{G}}}$. GATE calculates the variance ratio based on

30 randomly selected genotyped markers with MAC $\geq 20$ and computes the coefficient of variation (CV). If the CV of the variance ratios is smaller than 0.001, then the mean of the variance ratios is selected as $\hat{r}$, otherwise more markers are selected at an increment of 10 markers, and the CV is recalculated until the CV becomes smaller than 0.001.

## Score test using SPA

Using the estimated variance ratio $\hat{r}$, the variance-adjusted test statistic can be calculated as $T_{adj} = \widetilde{\mathbf{G}}(\boldsymbol{\delta} - \hat{\boldsymbol{\mu}}) / \sqrt{\hat{r}\widetilde{\mathbf{G}}\mathbf{W}\widetilde{\mathbf{G}}}$, under the null hypothesis has mean zero and variance unity. The traditional score test then assumes asymptotic normality of the score statistic $T$ (and thus $T_{adj}$ as well) under $H_0$, to calculate the $p$-value. However, observations have been made before in the context of logistic mixed models that the asymptotic normality assumption of the score test statistic leads to severe Type I error inflation for low-frequency and rare variants when the case-control ratio is unbalanced[19]. We make the same observations in frailty models as well when the censoring rate is high. In order to provide well-calibrated $p$-values in such situations, we used saddle point approximation (SPA) to approximate the null distribution of the score statistic, which has been shown to have better approximation error bounds compared to the normal approximation[39,41,71,72], especially at the extremely small tail probability region of $\alpha = 5 \times 10^{-8}$. Contrary to the normal approximation which only utilizes the first two moments only to approximate, SPA utilizes the entire moment generating function (MGF). In fact, it uses the cumulant generating function (CGF), i.e., is the logarithm of the MGF, which for the frailty model, based on the modified Poisson mixed model likelihood, can be derived as $K(\xi) = \sum_{i=1}^{N} \hat{\mu}_i(e^{\widetilde{G}_i c \xi} - \widetilde{G}_i c \xi - 1)$, where $c = \left(\hat{r}\widetilde{\mathbf{G}}\mathbf{W}\widetilde{\mathbf{G}}\right)^{-1/2}$. Then, the distribution of $T_{adj}$ can be calculated based on the SPA by $\Pr\left(T_{adj} < s\right) \approx \Phi\{w + \frac{1}{w}\log(\frac{v}{w})\}$, and the $p$-value is given by $p = \Pr\left(T_{adj} < -|s|\right) + \Pr\left(T_{adj} > |s|\right)$, where $T_{adj} = s$ is the observed adjusted score statistic, $w = \text{sign}\left(\hat{\xi}\right)\sqrt{2\left(\hat{\xi}s - K\left(\hat{\xi}\right)\right)}$, $v = \hat{t}\sqrt{K''\left(\hat{\xi}\right)}$, $\hat{\xi}$ is the solution to the equation $K'\left(\hat{\xi}\right) = s$, and $K'(\xi)$ and $K''(\xi)$ are the first and second derivatives of the CGF $K(\xi)$, respectively.

Since the normal approximation works well around the mean, we use the normal approximation when $T_{adj}$ is less than two standard deviations away from the mean for faster computation. In addition, a faster version of the SPA similar to Dey et al.[39] is also implemented which reduces the computation time even further, from $O(N)$ to $O(N_c)$, where $N_c$ is the number of minor allele carriers.

## Proportional hazard assumption

The proportional hazard (PH) assumption in frailty models is an extremely popular modeling assumption and has been widely used in biomedical research[2-4], as well as in GWAS[5-11]. In practice, diagnostics for the PH assumption[73-75] are difficult and time-consuming, and the PH assumption is thus impractical to be tested at such a large scale (both sample size-wise and genome-wide). To the best of our knowledge, no scalable diagnostic tool is available for testing proportional hazards of a continuous covariate in a frailty model. However, since millions of variants are tested in a GWAS, the quantile–quantile (QQ) plot works as a more practical alternative tool for model diagnostics. The QQ plot allows researchers to capture any unexpected conservativeness or anti-conservativeness of the $p$-values that may arise from the violation of model assumptions.

## Data simulation

We carried out a series of simulations to evaluate the performance of GATE, including the type I error rates and power. To evaluate whether GATE can control type I error rates in presence of sample relatedness, we randomly simulated a set of 1,000,000 base-pair "pseudo" sequences, in which variants are independent of each other. Alleles for each variant were randomly drawn from Binomial ($n = 2$, $p = $ MAF). Then we performed the gene-dropping[76] simulation using these sequences as founder haplotypes that were propagated through the pedigree of 10 family members shown in Supplementary Fig. 6. We simulated genotypes of 150,000 genetic variants with MAF $\geq 1\%$ for 5000 independent samples and 500 families based on the pedigree to estimate the GRM on-the-fly in Step 1 of GATE and genotypes of 1.9 million genetic variants with MAC;$\geq 20$ for association tests in Step 2. MAFs were randomly sampled from the MAF spectrum in UK Biobank imputation data as shown in Supplementary Fig. 8. For each subject $i$, the censoring time $T_{ci}$ was randomly selected from an exponential distribution with mean $1/\lambda_c$ and the underlying failure time $T_{fi}$ was generated from a frailty model with the underlying exponential hazard function $T_{fi} = \frac{-\log(U_i)}{\lambda\exp(\eta_i)}$, where $U_i \sim$ uniform $(0,1)$ and $\eta_i$ is the linear predictor. Under the null hypothesis of no genetic effects, $\eta_i = X_{1i}^{\top}\alpha + b_i$, where $X_1$ is a covariate that was randomly drawn from $N(0,1)$, $\alpha$ is the coefficient and is 0.5 and $b_i$ is the random effect simulated from $N(0, \tau\psi)$ with $\tau = 0.1$ and 0.25, respectively, which is the variance component parameter. The time for subject $i$ is $t_i = \min(T_{ci}, T_{fi})$ and $\delta_i = I(T_{fi} \leq T_{ci})$. We selected $\lambda$, the mean of the exponential hazard function, corresponding to different censoring rates $\sum_{i=1}^{N}\delta_i/N = 50\%$, 75% and 90%. We repeated the simulation 500 times. For each phenotype set, a null frailty model was fitted in Step 1 with the covariate $X_1$. In Step 2, we conducted single variant association tests on 1.9 million simulated genetic markers. In total, about $9.4 \times 10^8$ association tests were conducted. We evaluated the empirical type I error rates at the type I error rate $\alpha = 1 \times 10^{-6}$ and $5 \times 10^{-8}$ as shown in Supplementary Table 3 and Supplementary Fig. 7A. These results have indicated that GATE can produce well-calibrated type I error rates in the presence of sample relatedness at the significance level, while GATE-no SPA (similar to COXMEG) has inflated type I error rates and inflation gets larger than censoring rates is higher (Supplementary Table 3). For example, GATE-no SPA has type I error rate $8.9 \times 10^{-6}$ at $\alpha = 5 \times 10^{-8}$ when censoring rate is 75% and $2.8 \times 10^{-5}$ when the censoring rate is 90% with $\tau = 0.1$.

To evaluate whether GATE can control type I error rates in presence of cryptic sample relatedness, we have randomly selected $N = 10,000$ samples with white British ancestry from UK Biobank and simulated TTE phenotypes based on the observed genotyped of these subjects in the approach described above for pedigree-based data sets, except that under the null hypothesis of no genetic effects, $\eta_i = \boldsymbol{X}_{1i}^{\top}\boldsymbol{\alpha} + \sum_{j=1}^{L}\hat{G}_{ij}\beta$ and was simulated based on real genotypes of randomly selected $L = 30,000$ LD-pruned ($r^2 < 0.2$) markers from the odd chromosomes with MAF $\geq 1\%$. The real genotypes were used for simulating real sample relatedness in the null model. In particular, $X_1$ is a covariate that was randomly drawn from $N(0, 1)$, $\alpha$ is the coefficient and is 1, $\hat{G}_{ij}$ is the standardized genotype value for the jth marker of ith subject and $\beta$ is the genetic effect size following $N(0, \tau/L)$, where $\tau = 0.25$, which is the variance component parameter. The time for subject $i$ is $t_i = \min(T_{ci}, T_{fi})$ and $\delta_i = I\left(T_{fi} \leq T_{ci}\right)$. We selected $\lambda$, the mean of the exponential hazard function, corresponding to different censoring rates $\sum_{i=1}^{N}\delta_i/N = 50\%$, 75% and 90%. We repeated the simulation 100 times. For each phenotype set, a null frailty model was fitted in Step 1 with covariates including the first 4 genetic principal components, which were estimated for all White-British participants in the UK Biobank, and $X_1$. In Step 2, we conducted single variant association tests on genetic markers on the even chromosome. In total, $8.3 \times 10^8$ were conducted. We evaluated the empirical type I error rates at the type I error rate $\alpha = 1 \times 10^{-6}$ and $5 \times 10^{-8}$ as shown in Supplementary Table 4 and Supplementary Fig. 7B, which suggests that GATE produces well-calibrated type I error rates in the presence of cryptic relatedness at the corresponding significance levels.

To evaluate the empirical power of GATE and compare the power to COXMEG and SAIGE, phenotypes were generated under the alternative hypothesis for 10,000 samples, which contain 500 families and 5000 independent samples. The family pedigree is shown in Supplementary Fig. 6. We simulated the phenotypes for the $i$th individual under the alternative hypothesis $\beta \neq 0$ in the linear term $\eta_i = X_{1i}\alpha_1 + X_{2i}\alpha_2 + b_i + \sum_{j=1}^{10} \hat{G}_{ij}\beta$ of the underlying exponential hazard function for the underlying failure time $T_{fi} = \frac{-\log(U_i)}{\lambda \exp(\eta_i)}$, where $U_i$-uniform (0, 1), $G_{ij}$ is the genotype values for the $j$th marker, $\beta$ is the genetic log hazard ratio, $b_i$ is the random effect simulated from $N(0, \tau\psi)$ with $\tau = 0.25$. Two covariates, $X_1$ and $X_2$, were simulated from Bernoulli(0.5) and N(0, 1), respectively, with coefficients $\alpha_1$ and $\alpha_2 = 0.5$. $\lambda$ was determined to have a censoring rate of 50%. 100 datasets were simulated with 10 genetic markers with different hazard ratios. Power was evaluated at $\alpha = 5 \times 10^{-8}$ for MAF 0.05 and 0.2 as presented in Supplementary Figs. 9 and 10.

## UK Biobank TTE phenome

The time-to-event phenotypes for the UK Biobank were constructed as the disease phenotypes defined based on the hierarchical PheCodes[37] that represent different disease groups. The ICD9 and ICD10 codes were mapped to PheCodes using a combination of available maps through the Unified Medical Language System and other sources, string matching, and manual review[19,37]. For each PheCode, the subjects who had the PheCode were regarded as having failure events, and the subjects who did not have the PheCode were regarded as censored. For each failed subject, the TTE (failure time) was calculated by subtracting the birth year from the earliest time of diagnosis of any of the PheCode-specific ICD codes, rounded to the nearest full month. To obtain the TTE (censoring time) for each censored subject, the birth year was subtracted from the time of the last non-imaging visit to any of the UK Biobank ascertainment centers, or the last time any ICD code was recorded for that subject, or the time of death if death was recorded during the course of the study, whichever is latest, rounded to the nearest full month. For lifespan, the subjects who had their death recorded were assigned the failed status with the ages at death as the corresponding TTE, and the subjects who did not have their death recorded were assigned the censored status with the TTE defined as before.

## FinnGen

FinnGen is a public–private partnership project combining genotype data from Finnish biobanks and digital health record data from Finnish health registries (https://www.finngen.fi/en). Release 5 analysis contains 218,792 samples after quality control with population outliers excluded via principal component analysis based on genetic data. TTE phenotypes were constructed from population registries and ICD10 codes, and harmonizing definitions over ICD8 and ICD9, including ischemic heart disease ($N$ events = 30,952, $N$ censored = 187,838, censoring rate = 85.8%), female breast cancer ($N$ events = 8401, $N$ censored = 114,878, censoring rate = 93.2%), glaucoma ($N$ events = 8591, $N$ censored = 210,199, censoring rate = 96.1%) and Alzheimer's disease ($N$ events = 3899, $N$ censored = 207,324, censoring rate = 98.2%). We conducted genome-wide survival analysis using GATE with the first ten genetic PCs, sex, genotyping batch, and birth year as covariates and 240,000 pruned genetic markers for GRM estimation.

Patients and control subjects in FinnGen provided informed consent for biobank research, based on the Finnish Biobank Act. Alternatively, older research cohorts, collected prior to the start of FinnGen (in August 2017), were collected based on study-specific consent and later transferred to the Finnish biobanks after approval by Fimea, the National Supervisory Authority for Welfare and Health. Recruitment protocols followed the biobank protocols approved by Fimea. The Coordinating Ethics Committee of the Hospital District of Helsinki and Uusimaa (HUS) approved the FinnGen study protocol No. HUS/990/2017.

The FinnGen study is approved by Finnish Institute for Health and Welfare (THL), approval number THL/2031/6.02.00/2017, amendments THL/1101/5.05.00/2017, THL/341/6.02.00/2018, THL/2222/6.02.00/2018, THL/283/6.02.00/2019, THL/1721/5.05.00/2019, Digital and population data service agency VRK43431/2017-3, VRK/6909/2018-3, VRK/4415/2019-3 the Social Insurance Institution (KELA) KELA 58/522/2017, KELA 131/522/2018, KELA 70/522/2019, KELA 98/522/2019, and Statistics Finland TK-53-1041-17. The Biobank Access Decisions for FinnGen samples and data utilized in FinnGen Data Freeze 5 include: THL Biobank BB2017_55, BB2017_111, BB2018_19, BB_2018_34, BB_2018_67, BB2018_71, BB2019_7, BB2019_8, BB2019_26, Finnish Red Cross Blood Service Biobank 7.12.2017, Helsinki Biobank HUS/359/2017, Auria Biobank AB17-5154, Biobank Borealis of Northern Finland_2017_1013, Biobank of Eastern Finland 1186/2018, Finnish Clinical Biobank Tampere MH0004, Central Finland Biobank 1-2017, and Terveystalo Biobank STB 2018001.

## Genome build

The genomic coordinates reported in this paper were based on NCBI Build 37/UCSC hg19.

## Reporting summary

Further information on research design is available in the Nature Research Reporting Summary linked to this article.

## Data availability

UK Biobank

Individual-level genotype and phenotype data from the UK Biobank are available from http://www.ukbiobank.ac.uk. A formal application to the UK Biobank is required to download the data.

FinnGen

Individual-level genotype data from Finnish biobanks and digital health record data from Finnish health registries (https://www.finngen.fi/en) can be accessed from the Fingenious portal (https://site.fingenious.fi/en/). A formal approval for the researchers is required to access the data.

Availability of the GWAS results

The GWAS results for 871 time-to-event phenotypes in UK Biobank using GATE are currently available for public download at http://gate.genohub.org/. Manhattan plots, Q–Q plots, and regional association plots for each TTE phenotype as well as the PheWAS plots can be browsed at http://phewas.genohub.org/. The Registry of Open Data on AWS is accessed through https://registry.opendata.aws/broad-ukb-sumstats/.

## Code availability

GATE is implemented as an open-source R package available at https://github.com/weizhou0/GATE[77]. The FAVOR[61] portal is accessed through favor.genohub.org.

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

## Acknowledgements

The FinnGen project is funded by two grants from Business Finland (HUS 4685/31/2016 and UH 4386/31/2016) and 11 industry partners (AbbVie Inc, AstraZeneca UK Ltd, Biogen MA Inc., Celgene Corporation, Celgene International II Sàrl, Genentech Inc, Merck Sharp & Dohme Corp, Pfizer Inc., GlaxoSmithKline, Sanofi, Maze Therapeutics Inc., Janssen Biotech Inc.). Following biobanks are acknowledged for collecting the FinnGen project samples: Auria Biobank (www.auria.fi/biopankki), THL Biobank (www.thl.fi/biobank), Helsinki Biobank (www.helsinginbiopankki.fi), Biobank Borealis of Northern Finland (https://www.ppshp.fi/Tutkimus-ja-opetus/Biopankki/Pages/Biobank-Borealis-briefly-in-English.aspx), Finnish Clinical Biobank Tampere (www.tays.fi/en-US/Research_and_development/Finnish_Clinical_Biobank_Tampere), Biobank of Eastern Finland (www.ita-suomenbiopankki.fi/en), Central Finland Biobank (www.ksshp.fi/fi-FI/Potilaalle/Biopankki), Finnish Red Cross Blood Service Biobank (www.veripalvelu.fi/verenluovutus/biopankkitoiminta), and Terveystalo Biobank (www.terveystalo.com/fi/Yritystietoa/Terveystalo-Biopankki/Biopankki/). All Finnish Biobanks are members of BBMRI.fi infrastructure (www.bbmri.fi). This research has been conducted using the UK Biobank Resource under application number 52008. X.L. was supported by NCI R35-CA197449, P01-CA134294, U19-CA203654, and NHLBI R01-HL113338. B.M.N. was supported by NHGRI U01-HG009088-04S3 and NIMH R37-MH107649-06. R.D. was supported by NCI R35-CA197449. W.Z. was supported by an NIH T32 fellowship (Grant number: 1T32HG010464-01). A.P. was supported by the Academy of Finland Centre of Excellence in Complex Disease Genetics (Grant No. 312074). We would also like to acknowledge Cotton Seed, the Hail team, and the AWS Open Data Program (see the section "Data availability") for their help with data storage for UKBB summary statistics, and Hufeng Zhou and Theodore Arapoglou for their valuable help in setting up the website.

## Author contributions

R.D., W.Z., X.L., B.M.N., and M.J.D. designed experiments. R.D. and W.Z. performed experiments. R.D. and W.Z. implemented the software with input from X.L., B.M.N., and M.J.D. R.D. constructed phenotypes for UK Biobank data. R.D. and X.L. analyzed UK Biobank data. A.Q., R.D., and W.Z. created the PheWeb browser for UK Biobank results. W.Z., T.K., A.H., A.E., J.K., M.K., and A.P. analyzed data for the FinnGen study. Helpful advice was provided by S.L. R.D., and W.Z. wrote the manuscript with input from all co-authors.

## Competing interests

B.M.N. is on the Scientific Advisory Board of Deep Genomics, and is a consultant for CAMP4 Therapeutics, Takeda, and Biogen. X.L. is a consultant to AbbVie Pharmaceuticals and Verily Life Sciences. M.J.D. is a founder of Maze Therapeutics and on the scientific advisory board of BC Platforms. The remaining authors declare no competing interests.

## Additional information

## FinnGen

Wei Zhou [2,3,4,5,9], Tuomo Kiiskinen [5,6], Aki Havulinna[5,6], Amanda Elliott[1,2,3], Juha Karjalainen[2,3,4,5], Mitja Kurki[2,3,4,5], Aarno Palotie [2,3,4,5] & Mark Daly[2,3,4,5,10]

