## [Peer Review File · Nature Communications]

Efficient and accurate frailty model approach for genome-wide survival association analysis in large-scale biobanksEditorial Note: This manuscript has been previously reviewed at another journal that is not operating a transparent peer review scheme. This document only contains reviewer comments and rebuttal letters for versions considered at *Nature Communications* .

REVIEWER COMMENTS

Reviewer #1 (Remarks to the Author):

In the revised version, the authors addressed many of the previous comments raised in the original version of the manuscript. Nevertheless, several fundamental issues remain.

First, it is appreciated that the authors now included a new simulation study using COXMEG-sparse, in addition to COXMEG-score, to contrast the GATE model. The authors' benchmark shows, however, that the memory usage in COXMEG-sparse is about half of that in COXMEG-score. This high memory usage is puzzling because sparse matrices generally require much less memory. In order the researchers could replicate these benchmarks, the authors should disclose all relevant information about the sparsity of the GRM and the commands (or script) for running COXMEG-sparse.

The second is about SPA model in this manuscript. My apologies if the previous comment, "the analyses with such a small number of disease events are flawed," was unclear. The problem is that the analyses of variants with small MAC and a few disease events individually (i.e., when just one variant is included to the model) typically have very low power, regardless of whether SPA or any other method controls for the type I error rate. Then, the single-variant analysis in the case of small MAC and rare events will inevitably lead to a higher false discovery rate. This problem becomes evident in the real data analysis reported in this manuscript, in which almost no novel findings were replicated, as mentioned in the previous comment 6.1. The fact that the vast majority of these unreplicated findings are rare variants concerns the practical importance of the method in the case of small MAC and rare events. Then, researchers encouraged to use SPA for such variants have a big chance to select many false associations.

The third is the issue related to the models for time-to-event outcomes in humans. The authors correctly noted that most of the existing methods used in current GWA studies of time-to-event traits in humans employ the proportional hazards (PH) models, including the current model GATE, COXMEG, COXME, etc. However, as long as the researchers intend to use a PH model to analyze connections between genetic factors and complex (non-Mendelian) traits, they inevitably face the problems of proportionality of the hazards and their independence. This is because besides typical factors considered in the traditional GWA studies (such as those the authors emphasize, i.e., population structures and sample relatedness), these connections are affected by other multiple substantive factors inherent to non-Mendelian traits. For example, they include changes in gene functions with aging (i.e., senescence) and/or over time (e.g., secular trends), changes in physiological regulations with aging (e.g., blood pressure changes with aging) and over time (e.g., the epidemic of obesity in recent decades), the complexity of metabolic networks in an organism evolutionary adapted to increase fitness in various environments, changes in the developmental programs of an organism (e.g., growth cessation, menopause), epigenetic silencing and/or activation with aging and over time, gene-gene and gene-environment interactions, etc. Because these factors are inevitable in human populations and studies (including Biobank-size studies) usually include people from different birth cohorts and of different ages, the main problem in genetic association studies of complex traits is not just to have GWAS discoveries but to better understand the complex relationship between genetic factors and the complex traits.

The importance of this problem is clearly seen when considering the APOE e4 allele as an illustrative example. This is the missense (functional) variant, which is one of the best-studied variants in humans and one of the major "genetic discoveries." This allele provides the strongest contribution to the risk of

late-onset Alzheimer's disease as a single variant. There is no doubt that this is not a technical artifact, nor the result of population structure or sample relatedness bias. Nevertheless, despite nearly 28 years of research, this missense variant is considered a risk allele but not a causal variant, and its role in Alzheimer's disease remains elusive. This is the problem of the vast majority of GWAS discoveries for non-Mendelian traits (this problem is further complicated by smaller effect sizes than that for the e4 allele). This problem exists regardless of whether or not the analyses address population structures and sample relatedness. Increasing the sample size and computational speed in large-scale data just add new potential correlates of non-Mendelian traits, but does not address the main problem of better understanding of the complex relationship between genetic factors and these traits.

In addition, although GWAS often takes care of false positives, false negatives are another severe issue when using PH models. This issue arises because disproportional hazards and/or their non-independence bias the effect estimates. This means that without controlling for these two main constraints of the PH models, some results can be more significant compared to the logistic regression model, whereas the others are less significant. The issue of false negatives might be more severe than that of false positives because of: (i) missing promising associations due to complexity of the association signals with non-Mendelian traits (that, for example, contributing to the so-called "missing heritability" problem) and (ii) underestimation of the effect sizes (that is an essential issue for potential translation of genetic discoveries to health care).

Thus, the analyses using PH models provide just raw estimates, which should be further examined by testing whether or not the basic assumptions of the PH model are violated. The fact that GWA studies do not always perform the tests of the PH model assumptions does not imply that these tests should be ignored. The challenge is that these tests are time- and effort-consuming, and it is hard to apply them to large-scale data. Then, alternative methods can be used. For example, large-scale studies can prioritize variants based on, e.g., comparative analyses of the results from PH and logistic models, and then examine the associations with large differences between these models in more detail. In general, as long as there are no fast and accurate tests of the PH model assumptions and they are not ignored, the choice of the model(s) for the initial screening for potential associations is not of critical importance. This means that none of the PH models is a game-changer.

Given these considerations, although the proposed model is interesting, it represents incremental improvement in genetic association studies of complex, non-Mendelian traits (even if it will be clearer about potential gain in the computational efficiency).

The authors should also consider the following comments.

1. The authors' initiative to compare the results of the analyses using GATE (time-to-event model) and SAIGE (logistic model) is highly appreciated. However, Supplementary Table 3 provides just selected information from that comparison. Even so, this table shows the marginal decrease of p-values for several variants, and it does not show the effect sizes (betas). To accurately compare the results from these two models, the authors should present scatter plots for minus-log-base10 transformed p-values from GATE vs. the ones from SAIGE for unbiased results (i.e., similarly, as they did when contrasting GATE and COXMEG in Supplementary Figure 1). A similar scatter plot contrasting the effect sizes (betas) should complement the scatter plots for the p-values.

2. The authors reported that the p-value for the association of rs429358 with overall lifespan in FinnGen from GATE was eight orders of magnitude smaller (10^{-14}) than that in SAIGE (10^{-6}). The estimate of p-value in UK Biobank for this SNP was reported to be 10^{-5} (presumably, it is from GATE, given that Supplementary Table 4 reports hazard ratios). The authors did not report what the difference in p-values from GATE and SAIGE for rs429358 in UK Biobank was. Also, no such considerable improvements in p-values were reported for the other variants presented in Supplementary Table 3. Accordingly, the authors' claim that the huge difference in p-values rs429358 in FinnGen from GATE and SAIGE "demonstrated the significance of using frailty model to uncover genetic risk factors for TTE phenotypes" is not accurate because it is based just on one non-replicated observation in FinnGen. The observation of this huge difference is also puzzling as GATE does not fit the alternative model that affects the accuracy of the hazard ratio estimates. Please also see concern #1 above.

3. Abbreviations and notations in the Supplementary Tables should be spelled out.
4. Page 14, second paragraph from the top: it should be Supplementary Table 3.

Reviewer #2 (Remarks to the Author):

The authors have carefully and thoughtfully addressed all my previous comments. I would encourage another round of careful editing by the authors as I did see minor typos introduced in the revision process.

Reviewer #3 (Remarks to the Author):

The authors have made commendable effort to address the comments raised by reviewers for the previous version of this manuscript. In particular, I like the fact that now they have made some direct numerical comparison of results between logistic regression and cox model. However, I still have very significant concerns around this point. Below are my major comments that I would hope the authors can address.

1. Throughout the paper the authors emphasized the need for TTE model because of "heavy censoring" in cohort studies. In fact, the disease outcomes they analyze in the UK Biobank, fit into this "heavy censoring" scenario where a very small percentage (less than 5%) of the individuals develop the disease during the follow-up of the study. Yet in simulation (Supplemental Figure 10), where they compare power of TTE vs logistic model, they only consider much more common events which have 25% or 50% rates of occurrence during the follow-up. In fact, the simulations indicate as the disease become more rare, which means "heavier censoring", the difference in power between logistic regression and TTE diminishes. I would like the authors to report results from simulation studies where the event rates during the follow-up of the study is similar to the disease outcomes in the UK Biobank example. My anticipation is that with much smaller event rates, the difference between the TTE and logistic model will diminish and will become fairly negligible. I am happy to be proven incorrect.

2. I can see why for common outcomes, that has 50% or 25% rates, there could be significant difference in power between the two methods. Even if the time to disease onset indeed follows a Cox model, still the underlying hazard ratio (HR) parameters could be well approximated based on odds-ratio parameters of logistic models if the disease is rare. But if the disease is common and we force a logistic model to the data, then the logistic OR parameters will be substantially attenuated compared to the underlying HR parameters ---- this could lead to substantial loss of power. There is a lot of classic literature on connection between logistic regression and Cox model (see e.g Prentice and Breslow, *Biometrika*, 1978 but there is much more).

I think fitting actual TTE model has value to GWAS, but the logic the authors have is reversed. It might be more valuable for common outcomes than less common outcomes.

3. If my intuition is correct that there is expected to be less of a difference in power between logistic and TTE models for rare outcomes, such as the diseases the authors have analyzed in the UK Biobank, then I still remain very surprised by the relatively larger number of findings in the UK Biobank for the four rare disease by the proposed method. Here logistic regression based methods identified a relatively small number of loci, which seems expected as the number of cases/events is not very large. I would like to get a better insight to what is the reason there is such large differences in findings across the two approaches. If the simulation studies with disease rates similar to the four diseases observed in the UK Biobank do show that there is expected to be major power gain, then that

would be reassuring. Otherwise, the authors need to come up with some intuitive ways to explain the major differences across methods.

4. I appreciate that the author now spells out that main technical innovation here is to show how iteratively weighted Poisson GLM approach can be used for fitting frailty model under the null. There is a fairly large literature on computationally efficient method for fitting semiparametric frailty models. Given that for the score-tests, one needs to fit the null semiparametric frailty model only once, how much of an advantage it really is to be able use the Poisson GLM approach here. It would be helpful if the authors can really bring out the key advantage of the proposed method for fitting frailty model compared to alternative existing methods.

5. Discussion, Page 14, Second to last paragraph. The authors state "The TTE outcome is different from binary case-control outcome and logistic models are not appropriate for such outcomes". This is not necessarily true. As I indicated earlier, even when the underlying true model is a TTE model, one could get valid inference for hypothesis testing by fitting logistic regression model. The type-I error is typically still correct as the null value of the parameter means the same thing across the two models. There could be power loss, however, and substantially so when the outcome is more common. So I would suggest the authors rephrase what they mean by logistic regression being not appropriate for TTE outcomes.

Point-by-point Responses to the Reviewers' Comments

Title: An efficient and accurate frailty model approach for genome-wide survival association analysis controlling for population structure and relatedness in large-scale biobanks

Reviewer #1:

- *“First, it is appreciated that the authors now included a new simulation study using COXMEG-sparse, in addition to COXMEG-score, to contrast the GATE model. The authors’ benchmark shows, however, that the memory usage in COXMEG-sparse is about half of that in COXMEG-score. This high memory usage is puzzling because sparse matrices generally require much less memory. In order the researchers could replicate these benchmarks, the authors should disclose all relevant information about the sparsity of the GRM and the commands (or script) for running COXMEG-sparse.”*

Response: Thanks for the suggestion. We have included a section in the Supplementary Note (Section 8) to describe the computational resource requirement comparison between COXMEG and GATE. Specifically, the commands to run different methods, and the sparsity of the sparse GRMs in ‘dgCMatrix’ format are presented.

- *“The second is about SPA model in this manuscript. My apologies if the previous comment, “the analyses with such a small number of disease events are flawed,” was unclear. The problem is that the analyses of variants with small MAC and a few disease events individually (i.e., when just one variant is included to the model) typically have very low power, regardless of whether SPA or any other method controls for the type I error rate. Then, the single-variant analysis in the case of small MAC and rare events will inevitably lead to a higher false discovery rate. This problem becomes evident in the real data analysis reported in this manuscript, in which almost no novel findings were replicated, as mentioned in the previous comment 6.1. The fact that the vast majority of these unreplicated findings are rare variants concerns the practical importance of the method in the case of small MAC and rare events. Then, researchers encouraged to use SPA for such variants have a big chance to select many false associations.”*

Response: Thanks for the comment. Firstly, we agree that rare variants can have low power for phenotypes with few disease events. However, this does not mean that type I error/false discovery rate will be “inevitably higher”. There is no such relationship between power and type I error in the statistical literature. Moreover, we have shown through simulation studies that the type I error of GATE is well-controlled even for low MAF variants and heavily censored diseases. So, it is not true that type I error will be inflated regardless of the test. We hence respectfully disagree with the statement “Then, the single-variant analysis in the case of small MAC and rare events will inevitably lead to a higher false discovery rate.” Please excuse us if we have failed to understand this statement correctly.

Second, this is a method paper. The scope of our paper is to propose a method that we justify the validity through theoretical derivation, extensive simulation studies, and real-world examples of positive control variants. Novel discovery is not within the scope of this method paper, and we do not claim any of the significant associations to be novel. In our GWAS analysis of the four phenotypes from the UK Biobank data, 18 loci were identified by GATE which were missed by SAIGE. Twelve

out of these 18 loci have been previously found to be associated with the corresponding diseases in other studies, which suggests that these are positive control findings, and are not false positives. We do not claim the rest six to be true associations, and they could very well be false positives. No statistical test (except for the trivial test of always failing to reject) can guarantee that there will not be any false positives. However, we can quantify the chance of observing false positives through type I error rates, which have been shown to be well-controlled for GATE through our simulation studies.

Third, we respectfully disagree with the statement “Then, researchers encouraged to use SPA for such variants have a big chance to select many false associations.” The type I error is defined as the chance to select false associations under the null. We have shown that GATE with SPA controls type I errors correctly. Therefore, the claim that there would be a big chance to select false associations is not true. That chance is set at the type I error level of the test, which is 5×10^{-8} genome-wide, which is very low.

- *“The third is the issue related to the models for time-to-event outcomes in humans. The authors correctly noted that most of the existing methods used in current GWA studies of time-to-event traits in humans employ the proportional hazards (PH) models, including the current model GATE, COXMEG, COXME, etc. However, as long as the researchers intend to use a PH model to analyze connections between genetic factors and complex (non-Mendelian) traits, they inevitably face the problems of proportionality of the hazards and their independence. This is because besides typical factors considered in the traditional GWA studies (such as those the authors emphasize, i.e., population structures and sample relatedness), these connections are affected by other multiple substantive factors inherent to non-Mendelian traits. For example, they include changes in gene functions with aging (i.e., senescence) and/or over time (e.g., secular trends), changes in physiological regulations with aging (e.g., blood pressure changes with aging) and over time (e.g., the epidemic of obesity in recent decades), the complexity of metabolic networks in an organism evolutionary adapted to increase fitness in various environments, changes in the developmental programs of an organism (e.g., growth cessation, menopause), epigenetic silencing and/or activation with aging and over time, gene-gene and gene-environment interactions, etc. Because these factors are inevitable in human populations and studies (including Biobank-size studies) usually include people from different birth cohorts and of different ages, the main problem in genetic association studies of complex traits is not just to have GWAS discoveries but to better understand the complex relationship between genetic factors and the complex traits.*

The importance of this problem is clearly seen when considering the APOE e4 allele as an illustrative example. This is the missense (functional) variant, which is one of the best-studied variants in humans and one of the major “genetic discoveries.” This allele provides the strongest contribution to the risk of late-onset Alzheimer’s disease as a single variant. There is no doubt that this is not a technical artifact, nor the result of population structure or sample relatedness bias. Nevertheless, despite nearly 28 years of research, this missense variant is considered a risk allele but not a causal variant, and its role in Alzheimer’s disease remains elusive. This is the problem of the vast majority of GWAS discoveries for non-Mendelian traits (this problem is further complicated by smaller effect sizes than that for the e4 allele). This problem exists regardless of whether or not the analyses address population structures and sample relatedness. Increasing the sample size and computational speed in large-scale data just add new potential correlates of non-Mendelian traits, but does not address the main problem of better understanding of the complex relationship between genetic factors and these traits.

In addition, although GWAS often takes care of false positives, false negatives are another severe issue when using PH models. This issue arises because disproportional hazards and/or their non-independence bias the effect estimates. This means that without controlling for these two main constraints of the PH models, some results can be more significant compared to the logistic regression model, whereas the others are less significant. The issue of false negatives might be more severe than that of false positives because of: (i) missing promising associations due to complexity of the association signals with non-Mendelian traits (that, for example, contributing to the so-called “missing heritability” problem) and (ii) underestimation of the effect sizes (that is an essential issue for potential translation of genetic discoveries to health care).

Thus, the analyses using PH models provide just raw estimates, which should be further examined by testing whether or not the basic assumptions of the PH model are violated. The fact that GWA studies do not always perform the tests of the PH model assumptions does not imply that these tests should be ignored. The challenge is that these tests are time- and effort-consuming, and it is hard to apply them to large-scale data. Then, alternative methods can be used. For example, large-scale studies can prioritize variants based on, e.g., comparative analyses of the results from PH and logistic models, and then examine the associations with large differences between these models in more detail. In general, as long as there are no fast and accurate tests of the PH model assumptions and they are not ignored, the choice of the model(s) for the initial screening for potential associations is not of critical importance. This means that none of the PH models is a game-changer.

Given these considerations, although the proposed model is interesting, it represents incremental improvement in genetic association studies of complex, non-Mendelian traits (even if it will be clearer about potential gain in the computational efficiency).”

Response: Thanks for the comment. First, we agree that GWAS is not the end-all for genetic discovery and it is only the first step among many steps to unravel the complex genetic architecture of diseases. We agree that GWAS may not provide novel discoveries of causal variants. Our paper does not claim so either. Novel discoveries of causal variants is not within the scope of our paper. The purpose of GWAS is to find a set of candidate genetic markers out of millions of markers for potentially novel associations. These GWAS associations then need to be validated using independent samples. GWAS is an important first step of genetic discovery as evidenced by the extensive GWAS literature. To date, GWAS has identified tens of thousands of genetic variants and loci associated with a wide range of diseases and traits. Many large-scale biobanks, including the UK biobank, All of Us, Millions of Veteran Program (MVP), have been launched in recent years to accelerate GWAS discoveries.

These GWAS discoveries need to be investigated for their biological functions to determine whether they are causal variants for diseases and traits. We agree that functions of many of these GWAS discoveries are unknown and there is a substantial need to identify causal functional variants of these GWAS loci. Numerous large-scale efforts have been ongoing to study the functions of the variants identified by GWAS, and their translational values for drug target discovery and disease diagnosis and treatment. Examples include the recently launched NHGRI Impact of Genomic Variation on Function (IGVF) Consortium, Open Targets, a large-scale public-private partnership that uses human genetics and genomics data for systematic drug target identification and prioritization, and the International Common Disease Alliance (ICDA), which aims to improve prevention, diagnosis, and treatment of common diseases by accelerating discovery from genetic maps to biological mechanisms to

physiology and medicine.

The GATE method proposed in our paper improves over the existing methods and offers a computationally efficient and accurate method for scalable analysis of time-to-event outcomes in large biobanks (n=500,000 (UKB), n=1 million (AllOfUs) and n=1 million (MVP) that cannot be handled by the existing methods. Successful analysis of such large biobanks will accelerate GWAS discoveries and facilitate downstream variant biological functional and translational research efforts of several large consortia, such as IGVF, Open Targets, and ICDA.

Second, no statistical model can perfectly represent the true complexity of nature. As Dr. George Box said in his famous aphorism in statistics, “All models are wrong but some are useful.” The purpose of statistical modeling is to represent the complex reality using simple interpretable models with the help of practically reasonable model assumptions. The Proportional hazard (PH) assumption is by far the most popular modeling assumption for time-to-event outcomes as evidenced by over 57,000 citations to the Cox 1972 paper. It is true that in practice, diagnostics of the PH assumption is difficult and time-consuming, and is thus impractical to be tested for every variant in GWAS. However, the same holds true for the need to check the validity of logistic and linear mixed models that are widely used for analysis of any GWAS study. The lack of fast and accurate model assumption diagnostic methods does not make the method inapplicable. As a more practical alternative in GWAS, the quantile-quantile (QQ) plot is used as a very useful diagnostic tool to capture any unexpected conservativeness or anti-conservativeness of the p-values that may arise from the violation of model assumptions. Thus, we diligently present QQ plots for all the phenotypes including simulated phenotypes in our paper.

To address your comments, we have added discussions on functional and translational studies of GWAS discoveries to accelerate discovery from genetic maps to biological mechanisms to physiology and medicine, and identify and prioritize new drug targets (page 17, lines 497-505). We have also added discussions on model diagnostics in the Online Method Section (page 27-28, lines 700-710)

- *“1. The authors’ initiative to compare the results of the analyses using GATE (time-to-event model) and SAIGE (logistic model) is highly appreciated. However, Supplementary Table 3 provides just selected information from that comparison. Even so, this table shows the marginal decrease of p-values for several variants, and it does not show the effect sizes (betas). To accurately compare the results from these two models, the authors should present scatter plots for minus-log-base10 transformed p-values from GATE vs. the ones from SAIGE for unbiased results (i.e., similarly, as they did when contrasting GATE and COXMEG in Supplementary Figure 1). A similar scatter plot contrasting the effect sizes (betas) should complement the scatter plots for the p-values.”*

Response: Thanks for the suggestions. We have now included scatter plots for $-\log_{10}(p\text{-values})$ between GATE and SAIGE for the four example phenotypes in Supplementary Figure 11, and discussed the comparison in the Discussion Section (Page 14, lines 407-411). However, the meaning and interpretation of the effect sizes are very different between a frailty model and a logistic model. Hazard ratios and Odds ratios are different parameters, and only under very specific situations their interpretations become approximately similar. Specifically, for common events, regression coefficients in logistic regression models are attenuated compared to frailty models (Prentice and Breslow, *Biometrika*, 1978). These attenuated regression coefficients also result in power loss using logistic regression compared to survival analysis. Therefore, a direct comparison of the regression coefficient estimates of these two different models (Cox PH model and logistic mixed model) would

be inappropriate, and we refrain from presenting such a scatter plot.

- *“2. The authors reported that the p-value for the association of rs429358 with overall lifespan in FinnGen from GATE was eight orders of magnitude smaller (10⁻¹⁴) than that in SAIGE (10⁻⁶). The estimate of p-value in UK Biobank for this SNP was reported to be 10⁻⁵ (presumably, it is from GATE, given that Supplementary Table 4 reports hazard ratios). The authors did not report what the difference in p-values from GATE and SAIGE for rs429358 in UK Biobank was. Also, no such considerable improvements in p-values were reported for the other variants presented in Supplementary Table 3. Accordingly, the authors’ claim that the huge difference in p-values rs429358 in FinnGen from GATE and SAIGE “demonstrated the significance of using frailty model to uncover genetic risk factors for TTE phenotypes” is not accurate because it is based just on one non-replicated observation in FinnGen. The observation of this huge difference is also puzzling as GATE does not fit the alternative model that affects the accuracy of the hazard ratio estimates. Please also see concern #1 above.”*

Response: Thanks for the comment. First, we agree that a single variant data analysis example cannot be used to claim GATE outperforms SAIGE. That is why we performed extensive simulation studies to compare the performance of GATE and SAIGE. Our extensive simulation study results showed the potential improvement in power in GATE compared to SAIGE. The example of rs429358 in the GWAS of lifespan on FinnGen is an additional real-world example to support the observations already made through our simulation studies. It is not to be interpreted as the only evidence behind our claim. This example serves as a well-known positive control to show that GATE can identify loci that may be missed from a logistic model. To clarify the confusion, we have rephrased the sentence to “This example suggests that applying frailty models can be useful for uncovering genetic risk factors, as further evidenced through simulation studies (see ONLINE METHODS).” We have also included the p-value of rs429358 for lifespan on UK Biobank using SAIGE in Supplementary Table 4. Second, GATE is a score statistic-based test, and such tests are not impacted by the alternative model or accuracy of hazard ratio estimates. Score statistic-based tests such as GATE, do not require fitting an alternative model to correctly estimate the p-values.”

- *“3. Abbreviations and notations in the Supplementary Tables should be spelled out.”*

Response: Thanks for the suggestion. We have edited the supplementary tables accordingly.

- *“4. Page 14, second paragraph from the top: it should be Supplementary Table 3.”*

Response: Thanks for carefully pointing this out. We have corrected it now.

Reviewer #2:

“The authors have carefully and thoughtfully addressed all my previous comments. I would encourage another round of careful editing by the authors as I did see minor typos introduced

in the revision process.”

- Thanks for the encouraging remarks. We have carefully gone through the manuscript to correct the typos.

Reviewer #3:

“The authors have made commendable effort to address the comments raised by reviewers for the previous version of this manuscript. In particular, I like the fact that now they have made some direct numerical comparison of results between logistic regression and cox model. However, I still have very significant concerns around this point. Below are my major comments that I would hope the authors can address.”

- We thank the reviewer for the overall positive feedback.

1. Throughout the paper the authors emphasized the need for TTE model because of “heavy censoring” in cohort studies. In fact, the disease outcomes they analyze in the UK Biobank, fit into this “heavy censoring” scenario where a very small percentage (less than 5%) of the individuals develop the disease during the follow-up of the study. Yet in simulation (Supplemental Figure 10), where they compare power of TTE vs logistic model, they only consider much more common events which have 25% or 50% rates of occurrence during the follow-up. In fact, the simulations indicate as the disease become more rare, which means “heavier censoring”, the difference in power between logistic regression and TTE diminishes. I would like the authors to report results from simulation studies where the event rates during the follow-up of the study is similar to the disease outcomes in the UK Biobank example. My anticipation is that with much smaller event rates, the difference between the TTE and logistic model will diminish and will become fairly negligible. I am happy to be proven incorrect.”

Response: Thanks for the comment. First, we now have included a simulation study with 5% event rate for power comparison between GATE and SAIGE, and even though the power difference becomes smaller compared to the 25% and 50% scenarios, it is still non-ignorable. For instance, there is a ~5-6% power improvement in GATE compared to SAIGE at hazard ratio range ~2-3 for MAF 0.05, and at hazard ratio range ~1.5-1.8 for MAF 0.2. However, the more important thing is that GATE is not limited to be applied on rare diseases, it is perfectly valid to be applied to common diseases too, as shown in simulation studies. Providing valid inference in rare diseases is an added feature of GATE, and not the only feature. Rare diseases are overwhelmingly present in biobanks (see Supplementary Figure 2) and no survival analysis method exists to accurately analyze them. That is why we proposed a method that can analyze TTE phenotypes regardless of whether the phenotype is common or rare.

Furthermore, although the powers for more rare diseases are lower compared to common diseases, GATE provides accurate summary statistics for TTE phenotypes for individual biobanks, which can be input into meta-analysis of multiple biobanks to improve the power. In addition, given the UK biobank is a follow-up study, the participants have been followed over time, more events will be observed and the power will increase as the follow-up continues.

*“2. I can see why for common outcomes, that has 50% or 25% rates, there could be significant difference in power between the two methods. Even if the time to disease onset indeed follows a Cox model, still the underlying hazard ratio (HR) parameters could be well approximated based on odds-ratio parameters of logistic models if the disease is rare. But if the disease is common and we force a logistic model to the data, then the logistic OR parameters will be substantially attenuated compared to the underlying HR parameters ---- this could lead to substantial loss of power. There is a lot of classic literature on connection between logistic regression and Cox model (see e.g Prentice and Breslow, *Biometrika*, 1978 but there is much more).*

I think fitting actual TTE model has value to GWAS, but the logic the authors have is reversed. It might be more valuable for common outcomes than less common outcomes.”

Response: Thanks for the insightful and thoughtful remark. We apologize for the confusion, and completely agree with your assessment that the power gain using survival analysis over logistic regression will be more for common diseases than for rare diseases. Given the UK biobank participants will be followed over time, more events will be observed. We will observe more power gain using survival analysis compared to logistic regression as the followup continues. We have revised the manuscript to highlight this aspect (See page 3, line 50 and page 14-15, lines 413-418).

As shown in simulation studies, GATE provides valid inference for both common and rare outcomes. For 871 TTE phenotypes that have at least 500 events (cases) in the UKB data, 811 phenotypes have censoring rate more than 95%, and the lowest censoring rate among the 871 TTE phenotypes in the UK Biobank was ~77.4%. In view of a large number of TTE phenotypes in UKB are subject to heavy censoring, it is important that the proposed method can handle heavy censoring by properly controlling for type I error rate. We hence provided in the paper the UKB examples of heavily censored phenotypes, with the intention to highlight the need of the use of SPA in analyzing rarer phenotypes to control for type I error rate. We would like to note that even though the censoring rates are high for many TTE phenotypes, the numbers of disease events can be large because of the large sample sizes of biobanks. Furthermore, as the participants will be followed over time in biobanks, we will observe more events and TTE phenotypes will be more common, and the power gain of survival analysis over logistic models will be more, suggesting more benefits of using GATE over SAIGE for analysis of TTE phenotypes.

“3. If my intuition is correct that there is expected to be less of a difference in power between logistic and TTE models for rare outcomes, such as the diseases the authors have analyzed in the UK Biobank, then I still remain very surprised by the relatively larger number of findings in the UK Biobank for the four rare disease by the proposed method. Here logistic regression based methods identified a relatively small number of loci, which seems expected as the number of cases/events is not very large. I would like to get a better insight to what is the reason there is such large differences in findings across the two approaches. If the simulation studies with disease rates similar to the four diseases observed in the UK Biobank do show that there is expected to be major power gain, then that would be reassuring. Otherwise, the authors need to come up with some intuitive ways to explain the major differences across methods.”

Response: Thank you for the comment. In our analyses of the four example rare disease phenotypes, 114 loci were identified by GATE, out of which 18 loci (~16%) were not identified by SAIGE. We

would like to point out that 16% is not substantially larger than what we observe in the simulation studies. For instance, the empirical powers of GATE and SAIGE were 53.6% and 48.1% respectively for MAF 0.2 at the hazard ratio 1.6 with censoring rate 95%. This means, compared to GATE, SAIGE had ~10% less empirical power. As another instance, the empirical powers of GATE and SAIGE were 87.7% and 56.2% respectively for MAF 0.2 at hazard ratio 1.3 with censoring rate 75%. This means, compared to GATE, SAIGE had ~36% less empirical power. Therefore, failing to identify 16% of the SNPs by SAIGE that GATE identified in real data is not surprisingly different from the empirical observations based on the simulated data, especially considering that 11 out of the 18 loci were from the analysis of ischemic heart disease, which had a censoring rate of ~90.9%, in between 75% and 95%.

“4. I appreciate that the author now spells out that main technical innovation here is to show how iteratively weighted Poisson GLM approach can be used for fitting frailty model under the null. There is a fairly large literature on computationally efficient method for fitting semiparametric frailty models. Given that for the score-tests, one needs to fit the null semiparametric frailty model only once, how much of an advantage it really is to be able use the Poisson GLM approach here. It would be helpful if the authors can really bring out the key advantage of the proposed method for fitting frailty model compared to alternative existing methods.”

Response: Thank you for the comment. As discussed in the Introduction Section (2nd paragraph), most of the literature extensively studies the shared frailty model with a scalar random effect. However, the shared frailty model is extremely limited in scope to model complicated dependency structures such as family and cryptic relatedness among humans. Shared frailty models assume that the frailty (or the random effect) is the same for everyone in a cluster, which can only be true if subjects are genetically identical siblings (twins, triplets etc.). Bivariate extensions to the shared frailty model such as the correlated Gamma or correlated compound Poisson frailty model allow the frailties to be correlated among two subjects. However, these models are also too restrictive because they model the correlations using one parameter, and effectively, they are more appropriate for twin studies, and cannot model arbitrarily complex relationship structures.

The multivariate frailty model with Gaussian frailty is the only frailty model proposed in the literature, to the best of our knowledge, which can model complex relationship structures. COXME and COXMEG provides the commonly used approach to fit such a multivariate frailty model through the use of partial likelihood, similar to the approach of fitting Cox’s partial likelihood model. They are the most computationally efficient existing packages for fitting multivariate frailty models. We have shown that these two packages are not scalable for biobank-size data, and have compared GATE with these two procedures. GATE provides an alternative approach using a modified Poisson likelihood with a Breslow-type estimator used for the baseline hazards, similar to the approach that Breslow presented in the discussion of Cox 1972. Using modified Poisson GLMM approach allows us to use some efficient matrix operation techniques that were previously developed for GLMM fitting to make frailty model fitting scalable for biobank-size data, and also allows the application of the saddlepoint approximation to correct for the type I error inflation in the presence of heavy censoring.

In summary, the only existing methods that allows us to model arbitrarily complex relationship structures are COXME, COXMEG and GATE, and we have shown that GATE outperforms COXMEG (COXMEG outperforming COXME has been shown in He and Kulminski, 2020). We

would be happy to be suggested of any other frailty model which allows arbitrary covariance structures.

To respond to your comments, we have edited our literature review to add a short description of the existing correlated frailty models (and the relevant references are added in the paper) (See pages 3-4, lines 74-81).

“5. Discussion, Page 14, Second to last paragraph. The authors state “The TTE outcome is different from binary case-control outcome and logistic models are not appropriate for such outcomes”. This is not necessarily true. As I indicated earlier, even when the underlying true model is a TTE model, one could get valid inference for hypothesis testing by fitting logistic regression model. The type-I error is typically still correct as the null value of the parameter means the same thing across the two models. There could be power loss, however, and substantially so when the outcome is more common. So I would suggest the authors rephrase what they mean by logistic regression being not appropriate for TTE outcomes.”

Response: Thanks for the suggestion. We have rephrased the statement to, “The TTE outcome is different from binary case-control outcome and logistic models can result in loss of power for such outcomes.”

REVIEWER COMMENTS

Reviewer #2 (Remarks to the Author):

My comments are in the attached file.

Unfortunately, the revised version of this manuscript is not satisfactory because there are still concerns about two fundamental issues which the authors consider as the major advances of this manuscript.

The first issue is about the authors' claim about the efficiency of the GATE package, which the authors contrast to the COXMEG package.

First of all, I would like to thank the authors for including “a section in the Supplementary Note (Section 8) to describe the computational resource requirement comparison between COXMEG and GATE.” Although the authors claim that they used COXMEG-sparse in their comparative analysis as a benchmark, this new section shows that they incorrectly used COXMEG R package. Specifically, the second step with COXMEG-sparse still uses COXMEG-score rather than COXMEG-sparse because there is a line 'score=TRUE' in the code, as highlighted below.

8.4 Commands for running COXMEG-Sparse

```
#Load the dense GRM
load(paste0("pheno_coxmeg/sparseGRM.",N,".",rep,".RData"))
#Run the null model
re=coxmeg_plink(
  pheno=paste0("pheno_coxmeg/pheno.365.",N,".",rep,".txt"),
  corr=sparseGRM,
  spd=FALSE,
  type='sparse',
  cov_file=paste0("pheno_coxmeg/cov.365.",N,".",rep,".txt"),
  verbose=TRUE
)
#Run the association test
coxmeg_plink(
  pheno=paste0("pheno_coxmeg/pheno.365.",N,".",rep,".txt"),
  bed=paste0("geno/ukb_imp_chr21_v3_365.",N,".",rep),
  tmp_dir="coxmeg_temp",
  corr=sparseGRM,
  spd=FALSE,
  type='sparse',
  cov_file=paste0("pheno_coxmeg/cov.365.",N,".",rep,".txt"),
  tau=re$tau,
  score=TRUE,
  verbose=TRUE
)
```

This error explains why there is a trivial difference in the benchmark of Step 2, while a big difference is seen in Step 1 in Table 1, as highlighted below. This table confirms that the score test is implemented only for the dense matrix in COXMEG. However, the 'score=TRUE'

command will treat the GRM as a dense matrix. To implement the COXMEG package correctly for COXMEG-sparse, the authors should use 'score=FALSE' in Step 2, or exclude the 'score=TRUE' line from the code.

Table 1: Projected computation time and memory usage for GATE, COXMEG-Score, and COXMEG-Sparse across different sample sizes. Benchmarking was performed for the genome-wide association study (GWAS) of lifespan based on randomly subsampled data from UK Biobank White British ancestry subjects. Association tests (step 2) were performed on 200,000 randomly selected markers with imputation INFO ≥ 0.3 , with the filtering criteria of minor allele count (MAC) ≥ 20 , and the computation times were projected for testing 46 million variants. The reported run times are medians of five runs, each with randomly sampled subjects with different randomization seeds.

Method	Number of subjects	Step 1		Step 2	
		Time (hours)	Memory (GB)	Time (hours)	Memory (GB)
GATE	5000	0.114	0.36	26.635	0.25
	10000	0.707	0.54	30.730	0.24
	20000	0.488	0.74	33.794	0.24
	50000	1.190	1.44	57.579	0.28
	100000	6.081	2.91	80.059	0.34
	200000	11.592	5.55	135.217	0.46
	408582	31.285	10.60	287.135	0.87
COXMEG-Score	5000	0.029	1.80	191.625	1.70
	10000	0.118	7.99	737.053	6.02
	20000	0.555	32.75	3355.525	23.58
COXMEG-Sparse	5000	0.001	0.56	183.312	1.14
	10000	0.004	1.68	747.290	3.83
	20000	0.018	5.88	3096.988	14.40

The second issue is the same as in our previous version, i.e., about the analyses of variants with small minor allele count (MAC) and a few disease events individually, i.e., when just one variant is included in the model. The authors claim that they disagree with this comment, and they emphasize the type I error rate in their response.

Let's clarify that this issue is about the false discovery rate (FDR) but not the type I error rate (FPR). It is clear that SPA improves control of the FPR, but controlling FPR per se is not enough for the authors' claim that rare variants in rare diseases (or with few cases in the data) can be included in a single-variant analysis, even with improved control of FPR in the SPA. As the authors agree, rare variants in rare diseases/events have two problems, i.e., inflated FPR and low power. The SPA can handle the first issue only. This is, however, not sufficient for the validity of including such variants in the single-variant analysis because the second issue (which is not controlled by SPA) would lead to a large number of false discoveries. The problem that low power leads to a much higher FDR is well known. It is a bit surprising that the authors seem to be using FDR and FPR interchangeably ("type I error/false discovery rate").

To illustrate the problem, let's consider a toy example in which we test 100 SNPs, in which 5 are true signals, and the rest are true null. When FPR=5% and power=100%, we identify 5 true signals and 5 false positives that give FDR=5/10=50%. However, when the analysis is underpowered (e.g., assuming power=20%), even if we control the FPR (5%), we identify only one of the true signals and 5 false positives that give FDR=5/6=83.33%. Generally, for those variants that require the SPA to correct for the asymptotic normality of their score statistic, their

MAC is very low and, therefore, substantially underpowered. Including so many underpowered variants in the study gives a large number of false discoveries. This explains why almost none of the new findings of the rare variants are replicated in the authors' manuscript, as have been mentioned in the previous comment 6.1 in the first round of review. This is a good demonstration of the problem. If the authors want to control the FDR, they should use a more stringent threshold for p-value than $p=5e-8$, which would probably wipe out most of the findings. In fact, this is not a new issue. Over the last decade, a lot of efforts have been made to attempt to tackle this painful problem of power for rare variants by, for example, bundling them to increase the occurrence of minor alleles (e.g., the burden test and SKAT, just to name a few). Therefore, it can be dangerous to encourage geneticists to perform underpowered single-variant studies for rare variants with rare events by overblowing what the SPA can do and ignoring the FDR problems for rare variants.

Reviewer #3 (Remarks to the Author):

I would like to complement the authors again for a very thorough revision. I believe this revision has brought out the advantage of GATE more clear. I am also satisfied with the point-by-point response to my earlier comments. Overall this is a very fine paper that will significantly add to our toolbox for genetic analysis of large scale cohort studies.

Point-by-point Responses to the Reviewers' Comments

Title: An efficient and accurate frailty model approach for genome-wide survival association analysis controlling for population structure and relatedness in large-scale biobanks

Reviewer #2:

- *“The first issue is about the authors' claim about the efficiency of the GATE package, which the authors contrast to the COXMEG package. First of all, I would like to thank the authors for including “a section in the Supplementary Note (Section 8) to describe the computational resource requirement comparison between COXMEG and GATE.” Although the authors claim that they used COXMEG-sparse in their comparative analysis as a benchmark, this new section shows that they incorrectly used COXMEG R package. Specifically, the second step with COXMEG-sparse still uses COXMEG-score rather than COXMEG-sparse because there is a line 'score=TRUE' in the code, as highlighted below. This error explains why there is a trivial difference in the benchmark of Step 2, while a big difference is seen in Step 1 in Table 1, as highlighted below. This table confirms that the score test is implemented only for the dense matrix in COXMEG. However, the 'score=TRUE' command will treat the GRM as a dense matrix. To implement the COXMEG package correctly for COXMEG-sparse, the authors should use 'score=FALSE' in Step 2, or exclude the 'score=TRUE' line from the code.”*

Response: Thanks for going through our script meticulously and we apologize for the error. We have now fixed it as per your suggestion, and edited **Supplementary Note section 8, Supplementary Table 1, Figure 1**, and the **Computation and Memory costs** section in the main manuscript. Our results show that GATE remains substantially more efficient over COXMEG-Sparse in terms of computational costs: 98% and 88% reductions in computation time and memory (line 211-213).

- *“The second issue is the same as in our previous version, i.e., about the analyses of variants with small minor allele count (MAC) and a few disease events individually, i.e., when just one variant is included in the model. The authors claim that they disagree with this comment, and they emphasize the type I error rate in their response. Let's clarify that this issue is about the false discovery rate (FDR) but not the type I error rate (FPR). It is clear that SPA improves control of the FPR, but controlling FPR per se is not enough for the authors' claim that rare variants in rare diseases (or with few cases in the data) can be included in a single-variant analysis, even with improved control of FPR in the SPA. As the authors agree, rare variants in rare diseases/events have two problems, i.e., inflated FPR and low power. The SPA can handle the first issue only. This is, however, not sufficient for the validity of including such variants in the single-variant analysis because the second issue (which is not controlled by SPA) would lead to a large number of false discoveries. The problem that low power leads to a much higher FDR is well known. It is a bit surprising that the authors seem to be using FDR and FPR interchangeably (“type I error/false discovery rate”). To illustrate the problem, let's consider a toy example in which we test 100 SNPs, in which 5 are true signals, and the rest are true null. When FPR=5% and power=100%, we identify 5 true signals and 5 false positives that give FDR=5/10=50%. However, when the analysis is underpowered (e.g., assuming power=20%), even if we control the FPR (5%), we identify only one of the true signals and 5 false positives that give FDR=5/6=83.33%. Generally, for those variants that require the SPA to correct for the asymptotic normality of their score statistic, their MAC is very low and, therefore, substantially underpowered. Including so many underpowered variants in the study gives*

a large number of false discoveries. This explains why almost none of the new findings of the rare variants are replicated in the authors' manuscript, as have been mentioned in the previous comment 6.1 in the first round of review. This is a good demonstration of the problem. If the authors want to control the FDR, they should use a more stringent threshold for p-value than $p=5e-8$, which would probably wipe out most of the findings. In fact, this is not a new issue. Over the last decade, a lot of efforts have been made to attempt to tackle this painful problem of power for rare variants by, for example, bundling them to increase the occurrence of minor alleles (e.g., the burden test and SKAT, just to name a few). Therefore, it can be dangerous to encourage geneticists to perform underpowered single-variant studies for rare variants with rare events by overblowing what the SPA can do and ignoring the FDR problems for rare variants."

Response: Thanks for the helpful comments. First, we agree with you that single variant analysis has low power for testing rare variant effects, and SNP-set tests are needed for boosting the power of rare variant association tests. In the **Discussion section (pg 16-17, lines 473-482)**, we have expanded our discussions on this matter. We have provided more discussions on the limitation of single-variant analyses using GATE for rare variants, and noted that significant single variant based GWAS findings for rare variants need to be interpreted with caution, and replication of these findings using independent samples is important. We have discussed how to extend GATE to perform rare variant set-based tests such as SKAT or burden tests, using frailty models for censored time-to-event data in the future.

Second, we would like to note that throughout the manuscript, we have only used False positive rate (FPR, or Type I error) as our genome-wide significance criteria, as it is typically done in GWASs. We have never mentioned False Discovery Rate (FDR). The FDR control is beyond the scope of this paper. Given FDR is not commonly used in GWAS, it would be of future research interest to investigate FDR control in GWAS.

Finally, we would like to mention that SPA is very valuable even if one is interested in controlling FDR. Since FDR is a strictly increasing function of FPR, by bounding FPR for a given power value, the FDR will be bounded. Specifically, assuming SNPs are independent, we have $FDR = \left[1 + \frac{c\gamma}{(N-c)\alpha}\right]^{-1}$, where N is the number of SNPs tested, c is the number of causal SNPs, α is the FPR, and γ is the power¹. We demonstrate this through a toy example. Suppose, we test 100 million SNPs, out of which 100 SNPs are causal with 20% power. Then as SPA controls the FPR at $\alpha = 5 \times 10^{-8}$, the FDR using SPA will be ~ 20%. Whereas, without SPA, the FPR will be 5×10^{-5} (as seen through simulation studies, Supplementary Table 5), which results in FDR being ~ 99.6%. This toy example shows that by controlling FPR, SPA will greatly help controlling FDR as well.

Reviewer #3:

"I would like to complement the authors again for a very thorough revision. I believe this revision has brought out the advantage of GATE more clear. I am also satisfied with the point-by-point response to my earlier comments. Overall this is a very fine paper that will significantly add to our toolbox for genetic analysis of large scale cohort studies."

- Thank you for your positive comments. We appreciate your time and effort on reviewing our manuscript. Your comments and suggestions have certainly helped improve the manuscript.

References:

1. Storey, J.D. & Tibshirani, R. Statistical significance for genomewide studies. *Proceedings of the National Academy of Sciences* **100**, 9440-9445 (2003).

REVIEWERS' COMMENTS

Reviewer #2 (Remarks to the Author):

Thanks for addressing my comments.

I still have one more concern as I do not understand why computational time for GATE in Step 1 in your Table 1 in the supplemental text changes non-monotonically with increasing the sample size.

Point-by-point Responses to the Reviewers' Comments

Title: Efficient and accurate frailty model approach for genome-wide survival association analysis in large-scale biobanks

Reviewer #2:

- *“Thanks for addressing my comments. I still have one more concern as I do not understand why computational time for GATE in Step 1 in your Table 1 in the supplemental text changes non-monotonically with increasing the sample size.”*

Response: Thanks for the comment. We now have added an explanation in the first paragraph of Section 8.5 in the Supplementary Note which reads “In addition to the sample size, the computation time for GATE in step 1 depends on other factors such as the number of steps required for the pre-conditioned conjugate gradient (PCG) method to converge and estimation of the variance component, especially when the sample-size is small. This explains the non-monotonic nature of the median computation time for GATE step 1 as the sample size increases in the low sample size regime, however, the mean computation times are still monotonically increasing with sample size.” We have also included the mean computation times in addition to the median computation times previously reported in Supplementary Note Table 1.